# BE CONSISTENT! ENHANCING ROBUST VISUAL REASONING IN LVLMS WITH CONSISTENCY CONSTRAINTS

## ABSTRACT

While Large Vision-Language Models (LVLMs) exhibit strong perceptual capabilities, they remain vulnerable in visual reasoning tasks. Existing benchmarks largely focus on symbolic mathematical or scientific problems and simple vision-centric tasks, offering limited assessment of complex visual reasoning and logical consistency, a critical requirement for reliable reasoning systems. We introduce ConVBench, a complex vision-centric reasoning benchmark where each image is paired with two logically equivalent questions across six categories: action and state, complex counting, spatial reasoning, causal and intent understanding, commonsense reasoning, and temporal perception. To complement this benchmark, we define two evaluation metrics, logical consistency and robust accuracy, that jointly assess both correctness and consistency of model responses. We further present ConVLM, which improves LVLM reasoning through Group Relative Policy Optimization (GRPO)-based reinforcement learning with novel consistency reward. This method leverages automatically generated logically equivalent question–answer pairs and a dual reward design combining accuracy- and consistency-based signals, encouraging agreement between paired responses. The framework functions effectively with or without strict answer supervision. On ConVBench, ConVLM-7B achieves 73.36% logical consistency and 66.83% robust accuracy, setting a new state of the art among open-source models, and generalizes strongly to V*Bench (84.90% accuracy) and InfoVQA-test (81.90 ANLS).

## 1 INTRODUCTION

Although Large Vision-Language Models (LVLMs) (Liu et al., 2023) now achieve robust performance on visual perception tasks such as image captioning (Lin et al., 2015), their capacity for reliable visual reasoning remains far from resolved. Consequently, motivated by the rapid advances of Large Language Models (LLMs) on reasoning tasks, the multimodal community has increasingly turned to developing reasoning-centric LVLMs capable of complex inference grounded in visual evidence. In parallel, inspired by the success of Reinforcement Learning (RL) in enhancing reasoning for text-only models, many LVLM studies have adopted RL-based training. Notable examples include LLaVA-RLHF (Sun et al., 2023), which applies Reinforcement Learning with Human Feedback (RLHF) to vision–language alignment via factually augmented rewards, and FGAIF (Jing & Du, 2024), which trains fine-grained reward models from AI feedback and optimizes LVLMs through fine-grained feedback.

To evaluate the reasoning capabilities of LVLMs, the research community has introduced numerous mathematics- and science-oriented benchmarks (*e.g.,* MMBench (Liu et al., 2024b) and MathVista (Lu et al., 2024b)). However, these datasets typically involve limited visual reasoning; models often succeed by exploiting symbolic reasoning capabilities or domain-specific textual knowledge rather than faithfully interpreting visual evidence (Fu et al., 2024; Tong et al., 2024). To assess genuine visual competence, an alternative research direction proposes vision-centric benchmarks such as BLINK (Fu et al., 2024). Nevertheless, the visual problems in these benchmarks are typically superficial, often solvable "in the blink of an eye", thereby failing to evaluate complex reasoning processes. Furthermore, models frequently generate inconsistent rationales or answers for semantically equivalent queries, as illustrated in Figure 1, suggesting they exploit spurious correlations

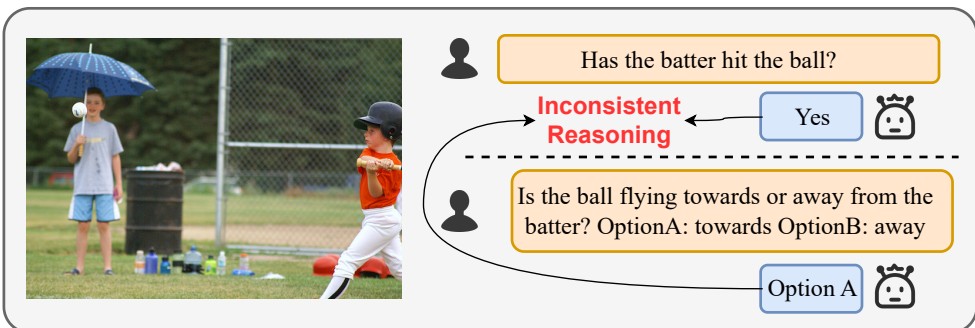

Figure 1: For the same image, an LVLM answers "Has the batter hit the ball?" with "Yes", but then predicts "towards" for "Is the ball flying towards or away from the batter?" If the batter has hit the ball, it should be moving *away*, so the two outputs are contradictory.

rather than demonstrating genuine understanding. Despite this critical limitation, the robustness of visual reasoning remains largely unexamined in current benchmarks.

To address these gaps, we introduce ConVBench, a complex vision-centric reasoning benchmark designed to evaluate both complex and consistent visual reasoning. Each image is paired with two logically equivalent questions whose answers should align (Figure 1). The benchmark spans six categories: action and state, complex counting, spatial relations, causal and intent understanding, commonsense reasoning, and temporal perception (Figure 2). To construct ConVBench at scale while ensuring quality, we adopt a two-stage pipeline: an LVLM first generates logically equivalent question–answer pairs, which are then validated by human annotators—accepted if correct, minimally edited if needed, or discarded otherwise. This procedure guarantees reliability without exhaustive manual authoring. Finally, to measure consistent reasoning, we propose two complementary metrics: logical consistency (whether paired answers agree) and robust accuracy (whether both are correct).

Despite recent progress in visual reasoning capabilities of LVLMs (Bai et al., 2025; Chen et al., 2024), our experiments reveal two critical limitations: models often generate *inconsistent* answers to semantically equivalent queries, and they still struggle with *complex, vision-centric* reasoning. Our goal is thus to develop a *robust* and consistent visual–reasoning model. Achieving this is non-trivial: deep models require substantial training data, and genuinely challenging visual reasoning supervision demands large quantities of carefully curated, high-quality annotations. A straightforward solution is to manually construct logically coherent question–answer pairs, but such labeling is costly and labor-intensive, making it infeasible to reach the scale and diversity needed for effective generalization.

To overcome this bottleneck, we introduce ConVLM, which combines *automatically generated* training pairs with a GRPO-based reinforcement learning objective (Shao et al., 2024). In particular, a proposer LVLM generates logically equivalent question pairs for each image and produces pseudo-answers. Our preliminary experiments show that while the generated questions are generally logically consistent, the pseudo-label answers are often noisy and frequently incorrect across both logically consistent questions. To mitigate this issue, we design a dual reward mechanism: alongside an accuracy reward, we introduce a consistency-based reward that does not rely on strict correctness but instead encourages agreement between answers to logically equivalent questions. This framework yields new state-of-the-art results among open-source models on ConVBench, surpassing even strong closed-source baselines such as Claude-3.7-Sonnet (Anthropic, 2025). Further evaluations on additional datasets demonstrate that ConVLM improves generalization rather than overfitting to our benchmark. Notably, even when trained solely with the consistency-based reward—*i.e.,* enforcing logical agreement without accuracy reward—ConVLM achieves substantial gains in both consistency and accuracy.

Our contributions are threefold: (i) We introduce ConVBench, the first benchmark that jointly evaluates complex vision-centric reasoning and logical consistency. (ii) We propose a weakly supervised consistency reinforcement framework, in which a proposer generates visual reasoning questions and

the model is optimized with both consistency- and accuracy-based rewards. (iii) Extensive experiments show that ConVLM achieves open-source state-of-the-art performance on ConVBench and exhibits strong cross-benchmark generalization.

## 2  CONVBENCH

We introduce ConVBench, a vision-centric benchmark specifically designed to rigorously evaluate LVLMs on complex reasoning. It spans six key categories: *Action and State* (AS), *Complex Counting* (CC), *Spatial Reasoning* (SR), *Causal and Intent* (CI), *Commonsense Reasoning* (CR), and *Temporal Perception* (TP), as shown in Figure 2. Each image is paired with two logically equivalent questions whose answers should align, enabling a direct assessment of reasoning consistency across models. We also compare our benchmark with the existing multimodal benchmarks in Appendix A.

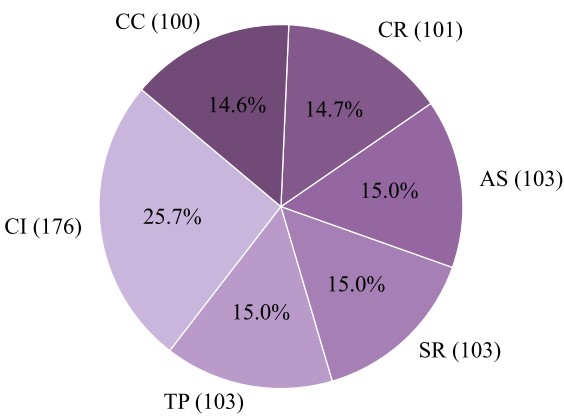

Figure 2: Question types in our ConVBench.

### 2.1  CATEGORIES

**Action and State**  This category centers on identifying the stage of an ongoing action, thereby assessing the model's ability to understand dynamic processes. To collect relevant images for annotation, we filtered candidates based on their captions, selecting those that contained action- or motion-related keywords. The complete keyword list is provided in Appendix K. To reduce human annotation costs, we employ large vision-language models to automatically generate logically equivalent question–answer pairs. These are created from prompts that incorporate the image, caption, bounding boxes, and task-specific instructions, as detailed in Appendix J. A similar automatic generation strategy is applied to the Complex Counting, Spatial Reasoning, Causal and Intent Reasoning, and Temporal Perception categories, with prompts tailored to the specific requirements of each task.

**Complex Counting**  This category addresses counting tasks under conditions of ambiguity, such as occlusion, mirrors, or reflections. It is intended to assess the model's capacity for disambiguation and accurate enumeration. For instance, when a person is standing in front of a mirror, the model should correctly infer that the reflection does not constitute an additional individual. To gather challenging images for this category, we filter the dataset based on image captions, selecting those containing keywords indicative of complex counting scenarios (*e.g.,* mirror). Comprehensive details and the full keyword list are provided in Appendix K.

**Spatial Reasoning**  This category aims to evaluate the model's ability to interpret relative positions and spatial arrangements of objects, thereby testing its spatial comprehension. To ensure the examples are sufficiently challenging, we filter images by object count, retaining only those that contain at least $m$ objects (see Appendix B).

**Causal and Intent Reasoning**  This category evaluates the model's ability to infer causes, intentions, and outcomes of actions, thereby assessing its capacity to reason about motivations and causal relationships. To identify images that are likely to involve causal reasoning, we filter the dataset using relevant keywords (*e.g.,* because). The complete keyword list used for filtering is provided in Appendix K.

**Commonsense Reasoning**  This category evaluates the model's ability to perform visual commonsense reasoning, requiring it to infer logically plausible situations and outcomes from visual

content. To leverage existing resources, we build upon established datasets such as VCR (Zellers et al., 2019) by extracting examples from its test split. For each selected example, we use the original question–answer pair as a reference and prompt a large vision–language model to generate a logically consistent complementary question–answer pair (see Appendix J for details). In this way, we construct our commonsense reasoning subset.

**Temporal Perception**   This category assesses the model's ability to understand and interpret temporal information from static visual inputs, with a focus on tasks such as reading clock faces and recognizing time-related contextual cues. For instance, when presented with an image of a clock, the model may be asked, "What time is shown?", which requires accurate perception of the clock hands as well as contextual signals indicating the time of day (e.g., daytime versus nighttime). To construct this category, we filter images whose captions contain time-related keywords (*e.g.,* clock). The complete keyword list is provided in Appendix K.

## 2.2 HUMAN CHECK

Because automatically generated question–answer pairs may contain factual inaccuracies, we conducted a comprehensive manual validation of every instance. Human annotators examined each question–answer pair against the corresponding visual input, applying the following criteria: pairs deemed correct and logically equivalent were retained; those with minor issues were corrected when feasible; and irreparable cases were discarded. This manual review served to validate the automatically generated annotations and enhance the overall reliability of the dataset. After this rigorous validation process, the final category-wise distribution of the curated dataset is shown in Figure 2.

## 3 METHOD

In this section, we first formalize the research problem and then present our proposed method, ConVLM, which strengthens robust visual reasoning in LVLMs with the consistency constraints.

## 3.1 PROBLEM FORMULATION

Let $\mathcal{I}$ be a set of images, and let a vision–language policy model $\pi_\theta(o \mid I, q)$ generate a textual solution $o$ for an image–query pair $(I, q)$. For robust visual reasoning, we require not only *answer accuracy* for each query, but also *logical consistency* across *logically equivalent* queries about the same image. Formally, for each $I$ we consider a set of queries $\mathcal{Q}(I)$ together with an equivalence relation $\sim$; denote by $\mathcal{G}(I) \subset \mathcal{Q}(I) \times \mathcal{Q}(I)$ the set of equivalent pairs $(q_1, q_2)$ with $q_1 \sim q_2$. Let $r_{acc}(I, q, o) \in [0, 1]$ be a verifiable accuracy reward for answering $(I, q)$, and let $r_{con}(o_1, o_2) \in \{0, 1\}$ be a binary consistency indicator between two solutions.

Our learning objective is to maximize expected accuracy while enforcing cross–query consistency:

$$\max_\theta \quad \mathbb{E}_{I \sim p(\mathcal{I}), q \sim p(\mathcal{Q}|I), o \sim \pi_\theta(\cdot|I,q)} \big[ r_{\text{acc}}(I, q, o) \big] \tag{1}$$

$$\text{s.t.} \quad \mathbb{E}_I \, \mathbb{E}_{(q_1,q_2) \sim \mathcal{G}(I)} \mathbb{E}_{\substack{o_1 \sim \pi_\theta(\cdot|I,q_1) \\ o_2 \sim \pi_\theta(\cdot|I,q_2)}} \big[ r_{con}(o_1, o_2) \big] \;\geq\; \tau, \tag{2}$$

where $\tau \in [0, 1]$ specifies the desired level of agreement across logically equivalent prompts.

In practice, we optimize the Lagrangian relaxation of Eqs. equation 1–equation 2:

$$\max_\theta \, \mathbb{E}_{I,q,o} \big[ r_{\text{acc}}(I, q, o) \big] \;+\; \gamma \, \mathbb{E}_{I,(q_1,q_2),o_1,o_2} \big[ r_{con}(o_1, o_2) \big], \tag{3}$$

where $\gamma > 0$ trades off accuracy and consistency. For more details, please see the next section.

## 3.2 CONVLM

We propose ConVLM (the **Con**sistency **V**ision–**L**anguage **M**odel), a weakly supervised framework that reinforces consistency in visual reasoning. ConVLM employs a proposer module to automatically generate visual reasoning questions and optimizes vision–language models using a combination of consistency-based and accuracy-based rewards, thereby eliminating the need for human annotation. An overview of the method is presented in Figure 3, which comprises two main stages: (i) Logically Equivalent Questions Generation and (ii) Consistency-based Reinforcement Learning.

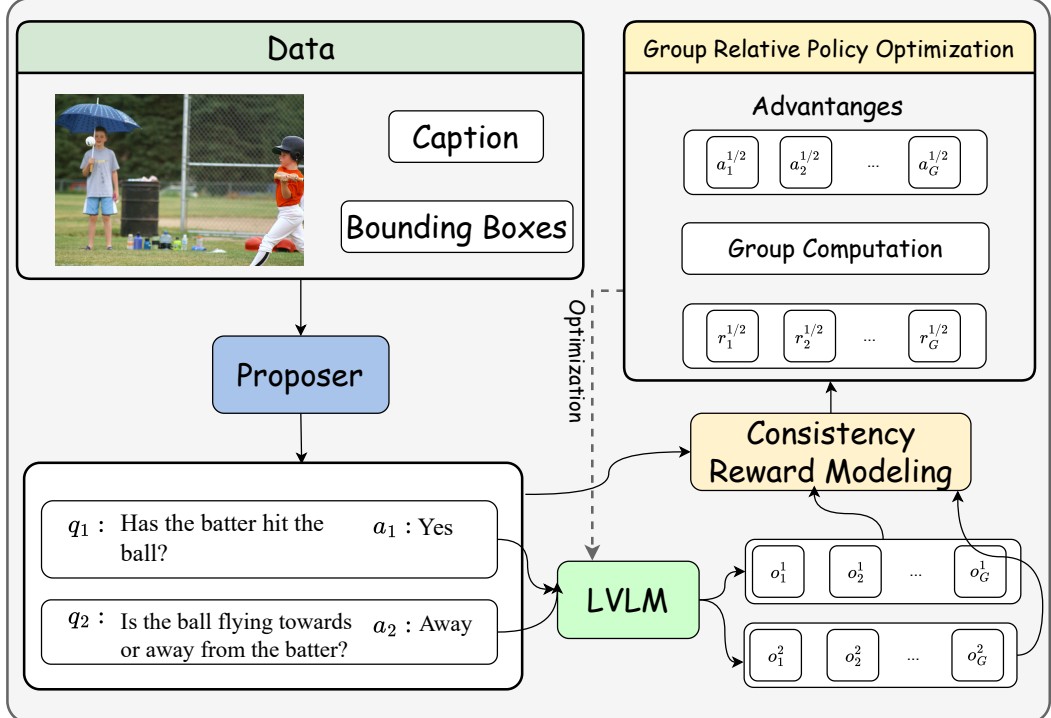

Figure 3: Pipeline of ConVLM. Given an image with its caption and bounding boxes, a proposer generates logically equivalent question pairs with corresponding pseudo-answers. The LVLM samples multiple candidate outputs for each question; a consistency reward evaluates agreement between paired solutions, while an accuracy reward assesses correctness against answer. GRPO subsequently aggregates these reward signals to optimize the model parameters. $a_i^{1/2}$ and $r_i^{1/2}$ represent the advantage and reward associated with the $i$-th generated answer to $q_1$ or $q_2$, respectively.

**Logically Equivalent Questions Generation.** A straightforward way to enhance a model's ability in consistent visual reasoning is to annotate logically equivalent question–answer pairs and train the model on them. However, manual annotation is costly and time-intensive. To overcome this limitation, we introduce a proposer model that automatically generates logically equivalent questions conditioned on the image, along with pseudo-answers that serve as supervision signals. Specifically, we employ GPT-4.1 (OpenAI, 2025) to generate paired logically equivalent questions and their corresponding answers based on the image, its caption, and object-level information:

$$\{(q_1, a_1), (q_2, a_2)\} = \mathcal{P}(P(I, C, O)), \tag{4}$$

where $I$, $C$, and $O$ denote the image, caption, and object information (including bounding boxes and categories), respectively, and $P(\cdot)$ is the prompt template defined in Appendix J. The output $(q_1, a_1), (q_2, a_2)$ represents the generated question–answer pairs, where $q_1$ and $q_2$ are logically equivalent. In other words, the answer to one can be inferred from the other, together with the visual context. Formally, this relationship is expressed as $a_1 \leftarrow f(I, q_2, a_2, q_1)$ or $a_2 \leftarrow f(I, q_1, a_1, q_2)$, where $f(\cdot)$ denotes the logical inference function.

**Consistency-based Reinforcement Learning** Recent advances increasingly adopt reinforcement learning to enhance reasoning capabilities in large models (Su et al., 2025; Zheng et al., 2023). Following this trend, we employ Group Relative Policy Optimization (GRPO) (Shao et al., 2024) to improve the complex visual reasoning ability of LVLMs. Originally introduced for mathematical reasoning in LLMs, GRPO can be effectively extended to visual reasoning tasks. Given a training input $(I, \mathbf{q})$, where $I$ denotes an image and $\mathbf{q}$ its associated query, GRPO optimizes the policy model using a reward function $r(I, \mathbf{q}, o)$. For each input, the old policy $\pi_{\theta_{old}}$ samples $n$ candidate responses, producing a reward set $\{r_i\}_{i=1}^n$. A baseline (mean reward) is computed, and the advantage of each

sample is normalized as:

$$\hat{A}_i = \frac{r_i - \text{mean}(r)}{\text{std}(r)}. \tag{5}$$

The GRPO objective adapts PPO (Schulman et al., 2017), constraining policy updates while maximizing the advantage-weighted clipped surrogate reward:

$$\mathcal{J}_{\text{GRPO}}(\theta) = \mathbb{E}_{(I,\mathbf{q}) \sim p_{\mathcal{D}}, \, \mathbf{o} \sim \pi_{\theta_{\text{old}}}(\cdot|I,\mathbf{q})}$$

$$\left[ \frac{1}{n} \sum_{i=1}^{n} \min \left( \frac{\pi_\theta(\mathbf{o}_i \mid I, \mathbf{q})}{\pi_{\theta_{\text{old}}}(\mathbf{o}_i \mid I, \mathbf{q})} \hat{A}_i, \, \text{clip}\left( \frac{\pi_\theta(\mathbf{o}_i \mid I, \mathbf{q})}{\pi_{\theta_{\text{old}}}(\mathbf{o}_i \mid I, \mathbf{q})}, \, 1 - \epsilon, \, 1 + \epsilon \right) \hat{A}_i \right) - \beta \mathbb{D}_{KL}\left( \pi_\theta || \pi_{ref} \right) \right], \tag{6}$$

where $\pi_\theta$ is the current policy, $\epsilon > 0$ sets the clipping threshold, and $\beta$ is a hyperparameter controlling the KL penalty.

A standard reward instantiation is a binary accuracy reward, which assigns 1 if the generated output contains the correct answer, and 0 otherwise (Shao et al., 2024):

$$r_{acc} = \begin{cases} 1 & \text{if the solution contains the correct answer to query,} \\ 0 & \text{otherwise.} \end{cases} \tag{7}$$

While straightforward, this reward relies on strict verifiability and can be brittle when reference answers are imperfect. Moreover, proposer-generated pseudo-answers may introduce noise. To address these issues, we introduce an additional consistency-based reward, which encourages agreement across logically equivalent queries. Although this signal does not guarantee correctness, it provides a complementary and noise-tolerant learning signal that drives the model toward logically consistent outputs, while the accuracy reward anchors correctness.

Specifically, we design two types of consistency-based rewards: a sample-wise reward and a group-wise reward, as illustrated in Figure 5. Let $\{o_1^1, \cdots, o_i^1, \cdots, o_G^1\}$ and $\{o_1^2, \cdots, o_i^2, \cdots, o_G^2\}$ denote the model-generated answers corresponding to the queries $q_1$ and $q_2$, respectively. Here, $o_i^1$ and $o_i^2$ represent the $i$-th generated solutions for $q_1$ and $q_2$. 1) For the sample-wise reward, which compares paired outputs directly, we take $o_i^1$ as an example and define the reward based on its consistency with $o_i^2$, as formulated below,

$$r_{con} = c(o_i^1, o_i^2), \tag{8}$$

where $c(\cdot)$ denotes the consistency function between $o_i^1$ and $o_i^2$. It returns 1 if the two outputs are logically consistent, and 0 otherwise. 2) For the group-wise reward, it compares one output against a group of responses from its equivalent query, as shown below,

$$r_{con} = \sum_{j=1}^{G} c(o_i^1, o_j^2)/G, \tag{9}$$

where $G$ is the number of solutions in the sampled group. Finally, we obtain our overall reward by summing the traditional reward and ours consistency-based reward as $r = r_{acc} + \gamma r_{con}$, where $\gamma$ is the hyperparameter that balances the accuracy reward and consistency-based reward.

**Discussion on Reward Hierarchy.** The reward for each solution $o$, *i.e.,* the combination of the accuracy reward $r_{acc}$ and the consistency-based reward $r_{con}$, leads to a natural stratification of the overall reward. Specifically, the highest reward is obtained when the generated outputs are both correct and consistent. In contrast, the lowest reward arises when the outputs are incorrect and inconsistent, *i.e.,* when the answer contradicts that to its logically equivalent question. Between these two extremes, there exist two intermediate situations: (i) correct but inconsistent, and (ii) incorrect but consistent. The relative contribution of these intermediate cases is governed by the hyperparameter $\gamma$, which adjusts the trade-off between correctness and consistency. Such a design yields a more diverse and discriminative reward signal, ensuring that the model is simultaneously encouraged to pursue correctness and maintain logical stability across equivalent queries.

## 4 EXPERIMENT

### 4.1 EXPERIMENTAL DETAILS

**Setups** We trained ConVLM on $8\times$ A100(40G) GPUs, using OpenRLHF (Hu et al., 2024) framework for reinforcement learning. For the RL training, we use a batch of 256. For the optimizer, we

Table 1: The performance comparison between different baselines and our ConVBench across six sub-tasks of our ConVBench.

| Model | Causal & Intent | | Temporal | | Spatial | | Action State | | Commonsense | | Complex Counting | | Average | |
|---|---|---|---|---|---|---|---|---|---|---|---|---|---|---|
| | Con. | Acc. | Con. | Acc. | Con. | Acc. | Con. | Acc. | Con. | Acc. | Con. | Acc. | Con. | Acc. |
| Close Source | | | | | | | | | | | | | | |
| Claude-3-5-sonnet-V2 | 61.93 | 48.30 | 65.05 | 46.60 | 67.96 | 55.33 | 66.02 | 53.39 | 48.51 | 27.72 | 64.00 | 44.00 | 62.24 | 45.89 |
| Claude-3-7-sonnet | 75.00 | 64.77 | 74.76 | 60.19 | 65.05 | 63.11 | 72.82 | 63.11 | 58.42 | 41.58 | 81.00 | 62.00 | 71.17 | 59.13 |
| GPT-4.1 | 84.09 | 78.41 | 79.61 | 77.67 | 80.58 | 73.79 | 71.84 | 69.90 | 59.41 | 54.46 | 85.00 | 72.00 | 76.76 | 71.04 |
| Gemini-2.5-Pro | 78.41 | 75.57 | 83.50 | 80.58 | 73.79 | 67.96 | 67.96 | 63.11 | 72.28 | 66.34 | 74.00 | 71.00 | 74.99 | 70.76 |
| Open Source | | | | | | | | | | | | | | |
| LLaVA-1.5-7B | 20.45 | 5.11 | 17.48 | 6.80 | 38.83 | 25.24 | 23.30 | 6.80 | 35.64 | 25.74 | 34.00 | 12.00 | 28.28 | 13.61 |
| LLaVA-1.5-13B | 34.09 | 23.30 | 42.72 | 33.98 | 31.07 | 18.45 | 46.60 | 38.83 | 16.83 | 6.93 | 35.00 | 21.00 | 34.38 | 23.74 |
| LLaVA-1.6-7B | 32.95 | 0.00 | 29.13 | 0.00 | 38.83 | 18.45 | 19.42 | 0.00 | 21.78 | 2.97 | 22.00 | 1.00 | 27.35 | 3.73 |
| LLaVA-1.6-13B | 41.48 | 0.57 | 30.10 | 4.85 | 33.01 | 7.77 | 23.30 | 0.97 | 24.75 | 2.97 | 23.00 | 4.00 | 29.27 | 3.52 |
| InternVL3-1B | 38.64 | 7.95 | 29.13 | 5.83 | 42.72 | 13.59 | 45.63 | 14.56 | 49.50 | 36.63 | 48.00 | 10.00 | 42.26 | 14.76 |
| InternVL3-2B | 43.75 | 19.32 | 45.63 | 27.18 | 54.37 | 23.30 | 39.81 | 10.68 | 59.41 | 44.55 | 56.00 | 28.00 | 49.82 | 25.50 |
| InternVL3-8B | 40.91 | 19.89 | 49.51 | 29.13 | 49.51 | 25.24 | 47.57 | 22.33 | 50.50 | 41.58 | 52.00 | 23.00 | 48.33 | 26.86 |
| InternVL3-9B | 53.98 | _41.48_ | 57.28 | 45.63 | 47.57 | 27.18 | 62.14 | _44.66_ | 58.42 | 46.53 | 51.00 | 35.00 | 55.06 | _40.08_ |
| InternVL3-14B | 55.11 | 32.39 | 52.43 | 30.10 | 56.31 | 36.89 | 51.46 | 27.18 | _62.38_ | _52.48_ | 59.00 | 28.00 | 56.11 | 34.50 |
| DeepSeek-VL-1.3b | 46.59 | 28.97 | 40.77 | 21.35 | 42.71 | 32.03 | 36.89 | 16.50 | 45.54 | 23.76 | 39.00 | 24.00 | 41.92 | 24.44 |
| DeepSeek-VL-7b | 51.70 | 33.52 | 57.28 | _49.51_ | 49.51 | _39.80_ | 44.66 | 37.86 | 59.40 | 38.61 | 46.00 | 37.00 | 51.42 | 39.38 |
| DeepSeek-VL2-tiny | _63.63_ | 0.56 | 59.22 | 0.00 | _71.84_ | 0.97 | 52.42 | 0.00 | 59.40 | 24.75 | _66.00_ | 0.00 | _62.08_ | 4.38 |
| DeepSeek-VL2-small | _63.63_ | 11.36 | _65.04_ | 6.79 | 58.25 | 10.67 | _64.07_ | 4.85 | 44.55 | 20.79 | 56.00 | 4.00 | 58.59 | 9.74 |
| Qwen2.5-VL-3B | 44.32 | 27.27 | 40.78 | 30.10 | 47.57 | 32.04 | 39.81 | 31.07 | 47.52 | 40.59 | 46.66 | _38.33_ | 41.89 | 33.23 |
| Qwen2.5-VL-7B | 55.11 | 38.64 | 60.19 | 41.75 | 41.75 | 11.65 | 50.49 | 38.83 | 56.44 | 36.63 | 56.00 | 28.99 | 53.32 | 32.75 |
| ConVLM-3B | 66.48 | 62.50 | 74.76 | 70.87 | **72.82** | **62.14** | 77.67 | 74.76 | 63.37 | **61.39** | 61.00 | 55.00 | 69.34 | 64.44 |
| ConVLM-7B | **75.56** | **69.88** | **77.66** | **72.81** | 66.01 | 60.19 | **80.58** | **75.72** | **65.34** | 60.39 | **75.00** | **62.00** | **73.36** | **66.83** |

use the AdamW (Loshchilov & Hutter, 2019) optimizer and the learning rate is set to $10e$-7. For each query, we sample 8 responses for GRPO (*i.e.,* $G$=8). $\beta$ is set to 0.0 to omit the KL divergence constraint following recent practices in (Liu et al., 2025). $\gamma$ is set to 0.5.

**Dataset** For the training stage, we randomly sampled 5,000 images from the training set of MSCOCO (Lin et al., 2015). For evaluation, we assess the model on ConVBench. For our benchmark, we define two kinds of new metrics to evaluate the performance of LVLMs, *i.e.,* consistency and robust accuracy. Specifically, consistency aims to evaluate the logical consistency (Cons.) between the generated answers to logically equivalent questions. We define the consistency indicator $C_I$ as 1 if both questions are answered either correctly or incorrectly, and 0 otherwise. Meanwhile, robust accuracy (Acc.) is designed to evaluate a model's strict correctness in answering logically equivalent questions. It requires the model to answer both questions correctly. The metric is formally defined as follows: $A_I = 1$ if both questions are answered correctly; otherwise, $A_I = 0$.

In addition, we include more visual reasoning benchmarks to test our model's generalization ability, including V*Bench (Wu & Xie, 2024) and InfoVQA-test (Mathew et al., 2022). The metrics of V*Bench and InfoVQA-test are accuracy and ANLS (Mathew et al., 2022), respectively.

**Baselines.** To verify the effectiveness of our ConVLM, we compare it with several baselines. We select open-source models LLaVA-1.5-7B (Liu et al., 2024a), LLaVA-1.5-13B, LLaVA-1.6-7B, LLaVA-1.6-13B, InternVL3-1B (Zhu et al., 2025), InternVL3-2B, InternVL3-8B, InternVL3-9B, InternVL3-14B, DeepSeek-VL-1.3B (Lu et al., 2024a), DeepSeek-VL-7B, DeepSeek-VL2-tiny (Wu et al., 2024), DeepSeek-VL2-tiny, Qwen2.5-VL-3B (Bai et al., 2025), and Qwen2.5-VL-7B. We also include closed-source LVLMs, including Claude-3.5-sonnet-V2 (Anthropic, 2024), Claude-3.7-sonnet (Anthropic, 2025), GPT-4.1 (Brown, 2020), and Gemini-2.5-pro (Comanici et al., 2025).

## 4.2 MAIN RESULTS

We evaluate ConVLM on ConVBench and report both *Consistency* and *Robust Accuracy* across six task families in Table 1. Our model establishes new state-of-the-art performance on both metrics. The 7B variant achieves 73.36% average Consistency and 66.83% Accuracy, while the more efficient 3B model attains 69.34% and 64.44%, respectively. Compared to the strong closed-source model Claude-3.5-Sonnet-V2 (62.24%/45.89%), our 7B model demonstrates substantial improvements of 11.12% in Consistency and 20.94% in Accuracy. Both model variants significantly outperform all

open-source baselines of comparable or larger scale, including InternVL3-14B (56.11%/34.50%) and Qwen2.5-VL-7B (41.89%/32.75%).

ConVLM-7B leads on core reasoning tasks: *Causal&Intent* 75.56%/69.88%, *Time Perceptron* 77.66%/72.81%, and *Action State* 80.58%/75.72%. It also tops *Commonsense* with 65.34%/60.39% and delivers the strongest results on the difficult *Complex Counting* task (75.00%/62.00%). Interestingly, in *Spatial* reasoning, the 3B model attains the highest Consistency (72.82%) while 7B keeps a strong Accuracy (60.19%), suggesting limited gains from scale on this subtask. Unlike several baselines that display relatively high Consistency but near–zero Accuracy on some tasks, ConVLM maintains *both* metrics at high levels, suggesting robust reasoning.

## 4.3 ABLATION

We perform ablation studies to validate our design choices, evaluating three variants: w/o-Acc (accuracy reward removed), w/o-Con (consistency reward reduced via $\gamma$=0.3), and w/o-Data (training data replaced with 5,000 random examples from the training set in (Su et al., 2025).). Results in Table 2 reveal distinct contributions from each component. The accuracy reward contributes 2.22%/2.70% on our benchmark and 4.70% on V*Bench, confirming its importance for general reasoning accuracy. The consistency reward provides 1.93%/4.18% improvements on our benchmark and 3.96% on V*Bench, with the notable finding that consistency training enhances both consistency and accuracy metrics. Our generated data yields the largest gains (19.45%/22.75% on our benchmark, 4.82% on V*Bench), indicating strong generalization beyond benchmark-specific optimization. The complete ConVLM framework achieves superior performance by effectively combining all components.

Table 2: Ablation study of our ConVLM-3B.

| Model | ConVBench | | V*Bench |
|---|---|---|---|
| | Con. | Acc. | Accuracy |
| ConVLM | 69.34 | 64.44 | 80.21 |
| w/o-Acc | 67.12 | 61.74 | 75.51 |
| w/o-Con | 67.41 | 60.26 | 76.25 |
| w/o-Data | 49.89 | 41.69 | 75.39 |
| Base | 41.89 | 33.23 | 64.39 |

## 4.4 ANALYSIS

**Generalization to Other Benchmarks** Table 3 shows strong generalization of our models to external visual reasoning benchmarks. On V*Bench, ConVLM-7B achieves 84.90% accuracy, exceeding all baselines including Gemini-2.5-Pro (+5.70%) and Qwen2.5-VL-7B (+14.50%). ConVLM-3B reaches 80.21%, outperforming LongLLaVA, Qwen2.5-VL-3B, and GPT-4o. On InfoVQA-test, ConVLM-7B attains competitive performance (81.90% ANLS) relative to closed models while surpassing open-source alternatives. ConVLM-3B similarly outperforms comparable baselines at 69.75%. These results demonstrate that our improvements generalize beyond benchmark-specific gains to broader visual reasoning tasks.

Table 3: Results across different visual reasoning benchmarks.

| Model | V* Bench | InfoVQA-test |
|---|---|---|
| | Accuracy | ANLS |
| **Close Source** | | |
| Gemini-2.0-Flash | 73.20 | 86.50 |
| Gemini-2.5-Pro | 79.20 | 84.00 |
| GPT-4o | 62.80 | 80.70 |
| **Open Source** | | |
| Video-R1 | 51.20 | 67.90 |
| LongLLaVA | 68.50 | 65.40 |
| Gemma3 | 62.30 | 59.40 |
| Qwen2.5-VL-3B | 64.39 | 64.26 |
| Qwen2.5-VL-7B | 70.40 | 80.70 |
| ConVLM-3B | 80.21 | 69.75 |
| ConVLM-7B | 84.90 | 81.90 |

**Group-wise Consistency vs. Sample-wise Consistency** We compare two approaches for implementing the consistency reward in our GRPO training, as described in the Method section. Sample-wise consistency (SampleCon) matches rollouts index-by-index across pairs of logically equivalent prompts, while group-wise consistency (GroupCon) computes average agreement against the counterpart set. Table 4 demonstrates that both variants substantially outperform the baseline (Qwen2.5-VL-3B: 41.89%/33.23% on our benchmark and 64.39% on

V*Bench). The GroupCon variant achieves 66.83% Consistency and 59.19% Accuracy on our benchmark (+24.94%/+25.96%) while improving V*Bench performance to 72.77% (+8.38%). The SampleCon variant yields slightly stronger gains, reaching 67.12%/61.74% on our benchmark (+25.23%/+28.51% percentage points) and 75.51% on V*Bench (+11.12%). Both formulations prove effective, with sample-wise consistency providing a marginally stronger learning signal while group-wise consistency remains highly competitive. Based on these results, we adopt the sample-wise consistency reward for our final model.

In addition, we compare different training strategies (SFT vs. GRPO) and analyze the impact of data scale on performance in Appendix C.

Table 4: Comparison across different consistency reward settings.

| Model | ConVBench | | V*Bench |
|---|---|---|---|
| | Con. | Acc. | Accuracy |
| Qwen2.5-VL-3B | 41.89 | 33.23 | 64.39 |
| w/-GroupCon | 66.83 | 59.19 | 72.77 |
| w/-SampleCon | 67.12 | 61.74 | 75.51 |

## 5 RELATED WORK

**Large Vision-Language Models** With the success of large-scale pre-training in language models such as LLaMA (Touvron et al., 2023) and vision foundation models (Dosovitskiy et al., 2021), researchers have linked pretrained image encoders with powerful LLM backbones to create Large Vision–Language Models (LVLMs) capable of understanding real-world images (Awadalla et al., 2023; Li et al., 2023a). Despite these advances, models trained purely with supervised fine-tuning (SFT) remain brittle, often failing on complex visual reasoning (Su et al., 2025). Motivated by the success of reinforcement learning (RL) in aligning textual LLMs with human preferences (Ouyang et al., 2022), recent work, *e.g.,* VReST (Zhang et al., 2025), MindOmni (Xiao et al., 2025), and TACO (Zheng et al., 2023), introduces verifiable reward or self-reward RL objectives to strengthen LVLM reasoning. Distinct from these single-question reward strategies, our approach proposes a *dual-question consistency reward* that jointly evaluates two logically linked queries about the same image, providing low-cost supervisory signals and yielding more robust visual reasoning.

**Evaluation in Large Vision-Language Models** A surge of specialised benchmarks now probes LVLMs)from low-level perception to high-level reasoning. Early "all-around" suites such as MME (Fu et al., 2023) and MMBench (Liu et al., 2024b) cover 14-plus vision–language subtasks and remain the de-facto yardsticks for holistic capability assessment, while LVLM-eHub (Xu et al., 2025) aggregates public datasets into a unified leaderboard for rapid model comparison. More recent efforts emphasize fine-grained diagnosis: SEED-Bench (Li et al., 2023b) introduces 19K multiple-choice queries spanning 12 evluation dimensions, and MM-Vet (Chen et al., 2023) designs complex cross-modal "multimodal veterinarian" tests that require integrated perception, recognition and reasoning. Task- or domain-centric benchmarks have also emerged. For example, LAMM (Yin et al., 2023) evaluates 2D/3D visual grounding, MathVista (Lu et al., 2024b) and M3Exam (Zhang et al., 2023) target mathematical or exam-style challenges in visual contexts, ReasonVQA (Tran et al., 2025) focuses on multi-hop commonsense reasoning, and a medical hallucination suite highlights safety in radiology VQA (Narayanan et al., 2024). By contrast, we construct a *consistent, complex, vision-centric reasoning benchmark* with paired logically equivalent questions.

## 6 CONCLUSION

In this paper, we introduced ConVBench, a vision-centric benchmark designed to jointly evaluate complexity and consistency in visual reasoning, and ConVLM, a vision-language model trained with a reinforcement learning approach that leverages automatically generated, logically equivalent question-answer pairs with a consistency reward. Across six reasoning categories, ConVLM establishes new state-of-the-art results among open-source models on ConVBench and demonstrates strong generalization to external visual reasoning benchmarks. Our analysis reveals three key insights: (i) consistency represents a meaningful dimension of robustness that is distinct from accuracy, (ii) a straightforward consistency reward yields substantial performance gains without requiring strict ground-truth matching, and (iii) GRPO-based training effectively complements supervised learning objectives while enhancing out-of-distribution transfer capabilities.

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

APPENDIX

# A  BENCHMARK COMPARISON

We present a comparison between our proposed benchmark and existing multimodal benchmarks in Table 5. The complex reasoning in our vision-centric benchmark targets reasoning that requires integrating multiple visual cues in real-world scenes and can't be resolved with a blink.

Table 5: Comparison with existing multimodal benchmarks.

| Benchmark | Images | Vision-Centric | Complex Reasoning | Logically Equivalent QA Pairs | Questions |
|---|---|---|---|---|---|
| BLINK (Fu et al., 2024) | 7,300 | ✓ | × | × | 3,800 |
| V* (Wu & Xie, 2024) | 191 | ✓ | × | × | 191 |
| CV-Bench (Tong et al., 2024) | - | ✓ | × | × | 2638 |
| MME (Fu et al., 2023) | 897 | × | × | × | 1,277 |
| MMBench (Liu et al., 2024b) | - | × | ✓ | × | 3,217 |
| InfoVQA (Mathew et al., 2022) | 5,485 | × | ✓ | × | 30,035 |
| MathVista (Lu et al., 2024b) | 5,487 | × | ✓ | × | 6,141 |
| M3Exam (Zhang et al., 2023) | - | × | ✓ | × | 12,317 |
| DocVQA (Mathew et al., 2021) | 12,767 | × | × | × | 50,000 |
| MM-Vet (Yu et al., 2024) | 200 | × | × | × | 218 |
| Seed-Bench (Li et al., 2023b) | - | × | × | × | 19,000 |
| MMMU (Yue et al., 2024) | - | × | ✓ | × | 11,500 |
| ChartQA (Masry et al., 2022) | 4,800 | × | ✓ | × | 9,600 |
| TextVQA (Singh et al., 2019) | 28,408 | × | × | × | 45,336 |
| OCRBench (Liu et al., 2024c) | 1,000 | × | × | × | 1,000 |
| ConVBench | 686 | ✓ | ✓ | ✓ | 1,372 |

# B  DATA CONSTRUCTION DETAILS

The data source (including images, captions, and object information) are from MSCOCO (Lin et al., 2015) except the commonsense category. $m$ is set to 5.

# C  MORE ANALYSIS

**Comparison between SFT and GRPO**  To compare training strategies and verify the effectiveness of our RL-based training method, we run three regimes under matched conditions: **RL** Using the same backbone (Qwen2.5-VL-3B) and training set, we (i) perform **SFT** to establish a strong supervised fine-tuning baseline, (ii) apply **RL** (GRPO) to test whether RL optimization improves robustness and generalization, and (iii) combine them in a two-stage **SFT+RL** pipeline to examine complementarity. We report results on ConVBench (in-distribution setting) and V*Bench (out-of-distribution setting) to isolate how each strategy affects accuracy and consistency in- and out-of-distribution. **SFT** alone boosts ConVBench from 41.89/33.23 to 72.55/67.04 (Consistency/Accuracy; +30.66/+33.81), but reduces V*Bench to 57.72 (−6.67 vs. base), suggesting overfitting to our benchmark. In contrast, our GRPO-based **RL** policy achieves 69.34/64.44 on ConVBench within 3 points of SFT, while boosting V*Bench to 80.21 (+22.49

Table 6: Performance comparison across combinations of training strategies.

| Model | ConVBench | | V*Bench |
|---|---|---|---|
| | Consistency | Accuracy | Accuracy |
| Qwen2.5-VL-3B | 41.89 | 33.23 | 64.39 |
| SFT | 72.55 | 67.04 | 57.72 |
| RL | 69.34 | 64.44 | 80.21 |
| SFT + RL | 73.78 | 67.94 | 76.04 |

over SFT; +15.82 over base), evidencing the effectiveness of reward–driven *exploration* for out-of-distribution generalization. Combining the two, **SFT+RL** reaches the best ConVBench scores 73.78/67.94 (+1.23/+0.90 over SFT) and a strong V*Bench 76.04, trading a small margin to RL (–4.17) for higher stability on ConVBench. To isolate the contribution of our reward design from any SFT-induced effects and to promote better generalization, our primary models are trained *exclusively* with GRPO-based reinforcement learning.

**Performance with Various Data Size**. To assess the impact of image data volume, we examined model performance across different dataset sizes. Specifically, we compared results for 1000, 2000, 3000, 4000, and 5000 training images, along with the untrained Qwen2.5-VL model, as shown in Figure 4. Both consistency and accuracy improve as the number of training images increases, with the most notable gains occurring between 0 and 2000 images. After 3000 images, performance growth slows, indicating diminishing returns from additional data. This suggests that while more image data generally benefits the model, there is a point beyond which further gains become marginal.

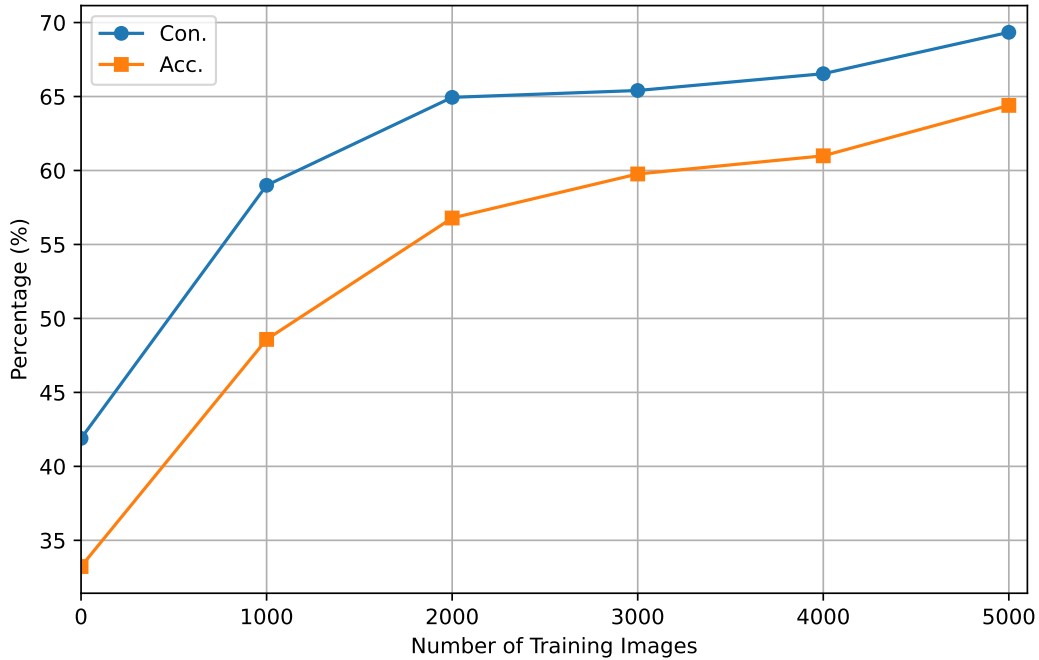

Figure 4: Model performance across different data sizes.

# D    FUTURE WORK

This work makes the initial step to explore consistency reward and robustness in visual reasoning tasks, leaving many promising directions to future work. 1) As an initial work, our formulation targets *pairwise answer consistency* for logically equivalent questions conditioned on a single image. We do not model *cross-image* consistency, *i.e.,* image pairs that depict the same visual semantics. 2) Moreover, we have not explored applying the proposed consistency reward to other modalities; extending it to *text-based reasoning* and *video-based reasoning*. 3) Furthermore, exploring improved consistency rewards, *e.g.,* task-specific variants or soft graded reward score (not just 0 or 1), is a promising avenue that could yield additional gains.

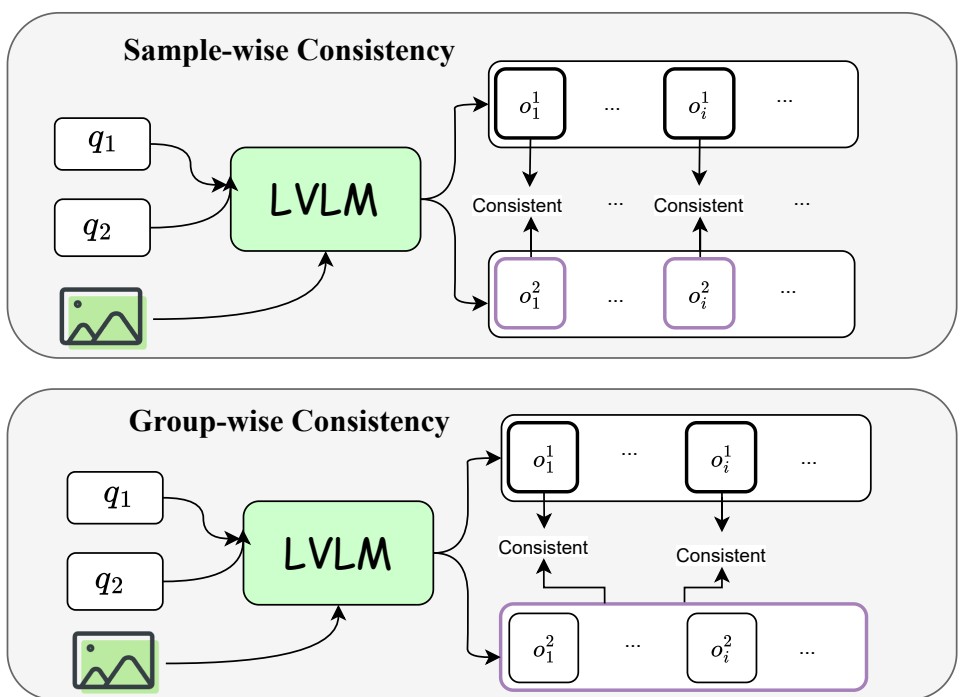

Figure 5: Visualization of sample-wise consistency reward and group-wise consistency reward.

## E    TRAINING PROCEDURE

For the consistency function $(c(\cdot))$, we implement it based on string matching. If both generated answers are correct or incorrectly, it is 1; otherwise, it is 0. To better illustrate the workflow of our method, we provided a simplified overview in Algorithm 1.

---
**Algorithm 1** Consistency-based Reinforcement Learning
---
**Input:** Initial policy model $\pi_{\theta_{init}}$; total training steps $M$; image set $\mathcal{I}$; hyperparameters $\gamma$, $\beta$
**Output:** Updated policy model $\pi_\theta$.

    Initialize policy model $\pi_\theta \leftarrow \pi_{\theta_{init}}$
    **for** step $= 1, \ldots, M$ **do**
        Sample a batch $\mathcal{I}_b$ from $\mathcal{I}$
        **for** each image $I \in \mathcal{I}_b$ **do**
            Generate logically equivalent question pair $\{(q_1, a_1), (q_2, a_2)\}$ for the image by Eqn.(4)
            Sample solutions $\{o_1^1, \cdots, o_G^1\} \sim \pi_\theta(\cdot \mid I, q_1)$ and $\{o_1^2, \cdots, o_G^2\} \sim \pi_\theta(\cdot \mid I, q_2)$
            Compute rewards for all sampled solutions
            Compute advantages by Eqn.(5)
        **end for**
        Update policy model by GRPO objective function in Eqn.(6)
    **end for**
---

## F    USE OF LARGE LANGUAGE MODELS

We used a large language model for copyediting. Specifically, we utilized OpenAI ChatGPT to suggest grammatical fixes, enhance wording and flow, and standardize terminology.

## G    ETHICS STATEMENT

Our work builds upon publicly available datasets and does not involve the collection of sensitive or private information. All human annotations were conducted with informed consent and fair compensation. Potential risks mainly relate to the misuse of improved reasoning capabilities in harmful applications; however, we emphasize that our benchmark and training framework are intended solely for research advancement

## H    REPRODUCIBILITY STATEMENT

To ensure the reproducibility of our work, we will release our datasets, source code, and the pretrained models. Detailed instructions and scripts will also be provided to facilitate replication of experiments and results reported in this paper.

## I    CASE STUDY

We show a qualitative comparison of several cases in Figure 6. In the first example (top row), two logically equivalent queries are posed about the same scene: "Are there more people on the sidewalk than cars on the road?" (GT: No) and "Is the number of people greater or less than the number of cars?" (GT: less). ConVLM-3B answers "No" and "less" consistently, whereas Qwen2.5-VL-3B answers "No" to the first but "greater (option A)" to the second, revealing reasoning inconsistency despite identical visual evidence. Similar observations can be observed in the remaining cases.

Cases 2–5 collectively reveal several recurring inconsistency patterns across different reasoning categories. Case 2 shows that intent and action understanding is fragile: small paraphrases around a player's pose (e.g., "signaling" vs. "ready to catch") can flip the prediction, indicating unstable grounding from fine-grained body posture to high-level intent. Case 3 highlights similar instability for dynamic outcomes, where a transitional frame (e.g., approaching a base or being tagged) causes the model to disagree on whether a goal state has been reached, even under logically equivalent descriptions. Case 4 exposes temporal reasoning issues: when asked about time-related properties (such as reading a clock or inferring "before/after" relations), the model can produce conflicting answers across equivalent formulations, suggesting it does not maintain a consistent internal representation of temporal state. Case 5 illustrates failures in causal and social reasoning, where different framings of the same situation lead to divergent explanations about motivations or consequences, despite the underlying visual evidence being unchanged. Together, these cases indicate that inconsistencies are most likely to occur when reasoning depends on subtle, multi-cue interpretations of actions, outcomes, time, and social context, underscoring the need for explicit consistency-oriented training.

## J    PROMPTS

### J.1    PROMPTS FOR DATA CONSTRUCTION

---

**Prompt for Causal and Intent**

**Image:**  ${image}$

**Prompt:**  You are a visual reasoning question generator. Given:
captions: a list of strings describing the same image
bboxes: a list of object entries in the form "$\langle category \rangle$ : [x, y, w, h]"
image: the input image contains object bounding boxes

Generate exactly one JSON object with four fields:
{ "direct_question": string, // a yes/no or simple fact question about an causal or intent reasoning in the image "direct_answer": string, // "Yes" or "No" (or a short fact) answering the direct question "direct_explanation": string, // Explain how can you get answer for the direct question, "indirect_question": string, // a multiple-choice question (two options) that

---

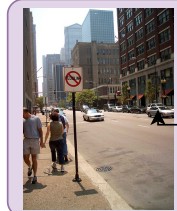

Q: Are there more people walking on the sidewalk than cars visible on the road?
GoundTruth: No ConVLM-3B: No   Qwen2.5-VL-3B: No

Q: Is the number of people walking on the sidewalk greater than or less than the number of cars visible in the scene? optionA: greater optionB: less
GoundTruth: optionB ConVLM-3B: optionB Qwen2.5-VL-3B: optionA

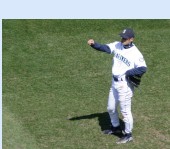

Q: Is the baseball player signaling with a raised arm?
GoundTruth: Yes ConVLM-3B: Yes   Qwen2.5-VL-3B: Yes

Q: Is the player in a ready stance to catch a ball or communicating with teammates? optionA: ready stance to catch optionB: communicating with teammates
GoundTruth: optionB ConVLM-3B: optionB Qwen2.5-VL-3B: optionA

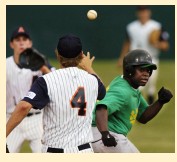

Q: Is the player in green (running) attempting to reach a base before being tagged out?
GoundTruth: Yes ConVLM-3B: Yes   Qwen2.5-VL-3B: No

Q: s the intent of the player in green to advance a base or to remain at his current base? optionA: advance optionB: remain
GoundTruth: optionA ConVLM-3B: optionA Qwen2.5-VL-3B: optionB

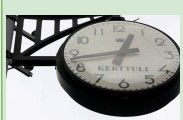

Q: Is it afternoon according to the clock?
GoundTruth: Yes ConVLM-3B: Yes   Qwen2.5-VL-3B: No

Q: Is the time shown on the clock closer to lunchtime or midnight?
optionA: lunchtime optionB: midnight
GoundTruth: optionA ConVLM-3B: optionA Qwen2.5-VL-3B: optionA

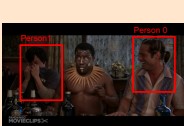

Q: Is the person on the left covering his face while smiling or laughing?
GoundTruth: Yes ConVLM-3B: Yes   Qwen2.5-VL-3B: No

Q: Why is person 1 covering his face? OptionA:  person 1 is taking cover from an explosion. OptionB: Because person 1 is laughing uncontrollably.
OptionC: Because person 1 is shocked about what has happened to person 0 .
OptionD: person 1 finds this process embarrassing .
GoundTruth: optionB ConVLM-3B: optionB Qwen2.5-VL-3B: optionB

Figure 6: Case study. The red fonts are the incorrect answer generated by models.

is logically implied by the direct question but phrased differently "indirect_answer": string // "optionA" or "optionB", "indirect_explanation":  string, // Explain how can you get answer for the indirect question }
Requirements: 1. Both questions must require reasoning about motion, sequence, posture or relative movement of objects/people. 2. The indirect question's correct choice must follow from the direct answer, but the wording/semantic focus should differ (e.g. "Did the batter hit the ball?" → "Is the ball moving toward or away from the batter? optionA: toward optionB: away"). 3. Use the bounding-boxes only to anchor who/what is acting—focus on actions, not

static attributes. 4. Do not invent objects or actions not supported by the captions + bboxes. 5. The object in explanations should appear with its coordinates if it appears in the image. 6. The explanation is used to train model. And note that the model will not be provided with captions and the bounding boxes. So please don't generate content like "the captions indicate xxx". 7. Direct questions are yes-or-no questions. Direct questions are asked in a direct way. Indirect questions are asked indirectly, from another angle. The answer to the indirect question can be inferred by combining the answer of direct question with the image. It is difficult to infer the answer to the indirect question only based on the answer to the direct question alone.

Focus on questions that involve causal reasoning (why something happened) or intent reasoning (what someone is trying to do).

Input Example captions: - "A young boy hits a baseball while another boy with an umbrella looks on." - "A little league player is swinging at a ball, and a boy under an umbrella is watching on the sideline." bboxes: - sports ball: [78.26, 108.53, 26.41, 26.97] - person: [63.74, 84.13, 104.0, 321.12] - baseball bat: [562.69, 203.18, 77.31, 22.34] - umbrella: [0.0, 8.33, 223.38, 96.05] - sports ball: [78.26, 108.53, 26.41, 26.97] ...

Expected Output Example "'json { "direct_question": "Has the batter already hit the ball?", "direct_answer": "Yes", "direct_explanation": "The batter's posture [63.74, 84.13, 104.0, 321.12] indicates a completed swing: the arms are fully extended, the bat [562.69, 203.18, 77.31, 22.34] has already passed through the hitting zone, and the ball [78.26, 108.53, 26.41, 26.97] is seen traveling away from the batter. These cues suggest the ball [78.26, 108.53, 26.41, 26.97] was just hit.", "indirect_question": "Is the ball flying toward or away from the batter? optionA: toward optionB: away", "indirect_answer": "optionB", "indirect_explanation": "The ball [78.26, 108.53, 26.41, 26.97] is flying away from the batter [63.74, 84.13, 104.0, 321.12], as the arms are fully extended, the bat [562.69, 203.18, 77.31, 22.34] has already passed through the hitting zone.", }

Please give me the output for the data. Captions: ${caption}$ Bounding Boxes: ${boudingbox}$ Output:

---

**Prompt for Complex Counting**

**Image:** ${image}$

**Prompt :** You are a visual reasoning question generator. Given:

captions: a list of strings describing the same image bboxes: a list of object entries in the form "⟨category⟩: [x, y, w, h]" image: the input image contains object bounding boxes

Generate exactly one JSON object with four fields:

{ "direct_question": string, // a yes/no or simple fact question about counting in the image "direct_answer": string, // "Yes" or "No" (or a short fact) answering the direct question "direct_explanation": string, // Explain how can you get answer for the direct question, "indirect_question": string, // a multiple-choice question (two options) that is logically implied by the direct question but phrased differently "indirect_answer": string // "optionA" or "optionB", "indirect_explanation": string, // Explain how can you get answer for the indirect question }

Requirements:

1. The indirect question's correct choice must follow from the direct answer, but the wording/semantic focus should differ. 2. Use the bounding-boxes to anchor content in the explanaiton. 3. Do not invent objects or actions not supported by the captions + bboxes. 4. The object in explanations should apear with its coordinates if it appears in the image. 5. The explanation is used to train model. And note that the model will not be provided with captions and the bounding boxes. So please don't generate content like "the captions indicate xxx". 6. Direct questions are yes-or-no questions. Direct questions are asked in a direct way. Indirect questions are asked indirectly, from another angle. The answer to the indirect question can be inferred by combining the answer of direct question with the image. It is difficult to infer the answer to the indirect question only based on the answer to the direct question alone.

Please generate complex counting questions that require reasoning, not just perception. Include situations like: - Counting objects with partial occlusion - Ignoring mirrored/reflected duplicates - Estimating counts in dense clusters

Input Example captions: - "A young boy hits a baseball while another boy with an umbrella looks on." - "A little league player is swinging at a ball, and a boy under an umbrella is watching on the sideline." bboxes: - sports ball: [78.26, 108.53, 26.41, 26.97] - person: [63.74, 84.13, 104.0, 321.12] - baseball bat: [562.69, 203.18, 77.31, 22.34] - umbrella: [0.0, 8.33, 223.38, 96.05] - sports ball: [78.26, 108.53, 26.41, 26.97] . . .

Expected Output Example "'json { "direct_question": "Has the batter already hit the ball?", "direct_answer": "Yes", "direct_explanation": "The batter's posture [63.74, 84.13, 104.0, 321.12] indicates a completed swing: the arms are fully extended, the bat [562.69, 203.18, 77.31, 22.34] has already passed through the hitting zone, and the ball [78.26, 108.53, 26.41, 26.97] is seen traveling away from the batter. These cues suggest the ball [78.26, 108.53, 26.41, 26.97] was just hit.", "indirect_question": "Is the ball flying toward or away from the batter? optionA: toward optionB: away", "indirect_answer": "optionB", "indirect_explanation": "The ball [78.26, 108.53, 26.41, 26.97] is flying away from the batter [63.74, 84.13, 104.0, 321.12], as the arms are fully extended, the bat [562.69, 203.18, 77.31, 22.34] has already passed through the hitting zone.", }

Please give me the output for the data. Captions: ${caption}$ Bounding Boxes: ${boudingbox}$ Output:

---

**Prompt for Action State**

**Image:** ${image}$

**Prompt:** You are a visual reasoning question generator. Given:

captions: a list of strings describing the same image bboxes: a list of object entries in the form "$\langle category \rangle$ : [x, y, w, h]" image: the input image contains object bounding boxes

Generate exactly one JSON object with four fields:

{ "direct_question": string, // a yes/no or simple fact question about an action or temporal relation in the image "direct_answer": string, // "Yes" or "No" (or a short fact) answering the direct question "direct_explanation": string, // Explain how can you get answer for the direct question, "indirect_question": string, // a multiple-choice question (two options) that is logically implied by the direct question but phrased differently "indirect_answer": string // "optionA" or "optionB", "indirect_explanation": string, // Explain how can you get answer for the indirect question }

Requirements: 1. Both questions must require reasoning about motion, sequence, posture or relative movement of objects/people. 2. The indirect question's correct choice must follow from the direct answer, but the wording/semantic focus should differ (e.g. "Did the batter hit the ball?" → "Is the ball moving toward or away from the batter? optionA: toward optionB: away"). 3. Use the bounding-boxes only to anchor who/what is acting—focus on actions, not static attributes. 4. Do not invent objects or actions not supported by the captions + bboxes. 5. The object in explanations should apear with its coordinates if it appears in the image. 6. The explanation is used to train model. And note that the model will not be provided with captions and the bounding boxes. So please don't generate content like "the captions indicate xxx". 7. Direct questions are yes-or-no questions. Direct questions are asked in a direct way. Indirect questions are asked indirectly, from another angle. The answer to the indirect question can be inferred by combining the answer of direct question with the image. It is difficult to infer the answer to the indirect question only based on the answer to the direct question alone.

Input Example captions: - "A young boy hits a baseball while another boy with an umbrella looks on." - "A little league player is swinging at a ball, and a boy under an umbrella is watching on the sideline." bboxes: - sports ball: [78.26, 108.53, 26.41, 26.97] - person: [63.74, 84.13, 104.0, 321.12] - baseball bat: [562.69, 203.18, 77.31, 22.34] - umbrella: [0.0, 8.33, 223.38, 96.05] - sports ball: [78.26, 108.53, 26.41, 26.97] . . .

Expected Output Example "'json { "direct_question": "Has the batter already hit the ball?", "direct_answer": "Yes", "direct_explanation": "The batter's posture [63.74, 84.13, 104.0, 321.12] indicates a completed swing: the arms are fully extended, the bat [562.69, 203.18, 77.31, 22.34] has already passed through the hitting zone, and the ball [78.26, 108.53, 26.41, 26.97] is seen traveling away from the batter. These cues suggest the ball [78.26, 108.53, 26.41, 26.97] was just hit.", "indirect_question": "Is the ball flying toward or away from the batter? optionA: toward optionB: away", "indirect_answer": "optionB", "indirect_explanation": "The ball [78.26, 108.53, 26.41, 26.97] is flying away from the batter [63.74, 84.13, 104.0, 321.12], as the arms are fully extended, the bat [562.69, 203.18, 77.31, 22.34] has already passed through the hitting zone.", } Please give me the output for the data. Captions: ${caption}$ Bounding Boxes: ${boudingbox}$ Output:

---

**Prompt for Time Perceptron**

**Image:** ${image}$

**Prompt:** You are a visual reasoning question generator. Given:
captions: a list of strings describing the same image bboxes: a list of object entries in the form "$\langle category \rangle$: [x, y, w, h]" image: the input image contains object bounding boxes
Generate exactly one JSON object with four fields:
{ "direct_question": string, // a yes/no or simple fact question about an temporal relation or time in the image "direct_answer": string, // "Yes" or "No" (or a short fact) answering the direct question "direct_explanation": string, // Explain how can you get answer for the direct question, "indirect_question": string, // a multiple-choice question (two options) that is logically implied by the direct question but phrased differently "indirect_answer": string // "optionA" or "optionB", "indirect_explanation": string, // Explain how can you get answer for the indirect question }
Requirements: 1. The indirect question's correct choice must follow from the direct answer, but the wording/semantic focus should differ. 2. Use the bounding-boxes to anchor content in the explanaiton. 3. Do not invent objects or actions not supported by the captions + bboxes. 4. The object in explanations should apear with its coordinates if it appears in the image. 5. The explanation is used to train model. And note that the model will not be provided with captions and the bounding boxes. So please don't generate content like "the captions indicate xxx". 6. Direct questions are yes-or-no questions. Direct questions are asked in a direct way. Indirect questions are asked indirectly, from another angle. The answer to the indirect question can be inferred by combining the answer of direct question with the image. It is difficult to infer the answer to the indirect question only based on the answer to the direct question alone.
These questions should focus on: - Understanding time from visual clues (e.g., clocks, lighting) - Reasoning about before/after events or what happens next
Input Example captions: - "A young boy hits a baseball while another boy with an umbrella looks on." - "A little league player is swinging at a ball, and a boy under an umbrella is watching on the sideline." bboxes: - sports ball: [78.26, 108.53, 26.41, 26.97] - person: [63.74, 84.13, 104.0, 321.12] - baseball bat: [562.69, 203.18, 77.31, 22.34] - umbrella: [0.0, 8.33, 223.38, 96.05] - sports ball: [78.26, 108.53, 26.41, 26.97] . . .
Expected Output Example "'json { "direct_question": "Has the batter already hit the ball?", "direct_answer": "Yes", "direct_explanation": "The batter's posture [63.74, 84.13, 104.0, 321.12] indicates a completed swing: the arms are fully extended, the bat [562.69, 203.18, 77.31, 22.34] has already passed through the hitting zone, and the ball [78.26, 108.53, 26.41, 26.97] is seen traveling away from the batter. These cues suggest the ball [78.26, 108.53, 26.41, 26.97] was just hit.", "indirect_question": "Is the ball flying toward or away from the batter? optionA: toward optionB: away", "indirect_answer": "optionB", "indirect_explanation": "The ball [78.26, 108.53, 26.41, 26.97] is flying away from the batter [63.74, 84.13, 104.0, 321.12], as the arms are fully extended, the bat [562.69, 203.18, 77.31, 22.34] has already passed through the hitting zone.", }

Please give me the output for the data. Captions: ${caption}$ Bounding Boxes: ${boudingbox}$ Output:

---

Prompt for Spatial Reasoning

**Image:** ${image}$

**Prompt:** You are a spatial reasoning question generator. Given:
captions: a list of strings describing the same image bboxes: a list of object entries in the form "$\langle category \rangle$: [x, y, w, h]" image: the input image contains object bounding boxes
Judge relative positions and spatial layouts of objects
Generate exactly one JSON object with four fields:
{ "direct_question": string, // a yes/no question about spatial relation in the image "direct_answer": string, // "Yes" or "No" (or a short fact) answering the direct question "direct_explanation": string, // Explain how can you get answer for the direct question, "indirect_question": string, // a multiple-choice question (two options) that is logically implied by the direct question but phrased differently "indirect_answer": string // "optionA" or "optionB", "indirect_explanation": string, // Explain how can you get answer for the indirect question }
Requirements: 1. The indirect question's correct choice must follow from the direct answer, but the wording/semantic focus should differ. 2. Use the bounding-boxes to anchor content in the explanaiton. 3. Do not invent objects or actions not supported by the captions + bboxes. 4. The object in explanations should apear with its coordinates if it appears in the image. 5. The explanation is used to train model. And note that the model will not be provided with captions and the bounding boxes. So please don't generate content like "the captions indicate xxx". 6. Direct questions are yes-or-no questions. Direct questions are asked in a direct way. Indirect questions are asked indirectly, from another angle. The answer to the indirect question can be inferred by combining the answer of direct question with the image. It is difficult to infer the answer to the indirect question only based on the answer to the direct question alone.
The questions should not be simple pairwise spatial relations. Instead, include more complex spatial reasoning types, such as: • multi-object reasoning (relations involving 3 or more objects) • indirect spatial relations (e.g., object A is to the left of object B, which is behind object C) • comparative spatial reasoning (e.g., which object is closer to X, or which is furthest from Y) • nested referential relations (e.g., object A is inside object B, which is next to object C) • ambiguous relations that require resolving occlusions, perspectives, or relative distances • vague spatial terms (e.g., near, far, adjacent to, close to) • ordering tasks (e.g., sort objects from left to right, or from nearest to furthest)
Avoid generating questions that are too trivial or directly answerable by simple object detection. Make sure the question requires actual spatial reasoning based on the scene context and multiple object relations.
Input Example captions: - "A young boy hits a baseball while another boy with an umbrella looks on." - "A little league player is swinging at a ball, and a boy under an umbrella is watching on the sideline." bboxes: - sports ball: [78.26, 108.53, 26.41, 26.97] - person: [63.74, 84.13, 104.0, 321.12] - baseball bat: [562.69, 203.18, 77.31, 22.34] - umbrella: [0.0, 8.33, 223.38, 96.05] - sports ball: [78.26, 108.53, 26.41, 26.97] ...
Expected Output Example "'json { "direct_question": "Has the batter already hit the ball?", "direct_answer": "Yes", "direct_explanation": "The batter's posture [63.74, 84.13, 104.0, 321.12] indicates a completed swing: the arms are fully extended, the bat [562.69, 203.18, 77.31, 22.34] has already passed through the hitting zone, and the ball [78.26, 108.53, 26.41, 26.97] is seen traveling away from the batter. These cues suggest the ball [78.26, 108.53, 26.41, 26.97] was just hit.", "indirect_question": "Is the ball flying toward or away from the batter? optionA: toward optionB: away", "indirect_answer": "optionB", "indirect_explanation": "The ball [78.26, 108.53, 26.41, 26.97] is flying away from the batter [63.74, 84.13, 104.0, 321.12], as the arms are fully extended, the bat [562.69, 203.18, 77.31, 22.34] has already passed through the hitting zone.", }

Please give me the output for the data. Captions: ${caption}$ Bounding Boxes: ${boudingbox}$ Output:

---

**Prompt for Commonsense**

**Image:** ${image}$

**Prompt:** You are a visual reasoning question generator. Given: Indirect Question: A commensence reseaning indirect questino for the image indirect_answer: the ansser for the indirect quesiont Image: the image indirect_explanation: explanation for why the answer can answer the question
Please generate a Yes-or-No direct-question, answer, explannaiton for the in-direct question, answer, and explanation.
Generate exactly one JSON object with four fields:
{ "direct_question": string, // a yes/no or simple question about an commonsense reasoning in the image "direct_answer": string, // "Yes" or "No" answering the direct question "direct_explanation": string, // Explain how can you get answer for the direct question, }
Direct questions are yes-or-no questions. Direct questions are asked in a direct way. Indirect questions are asked indirectly, from another angle. The answer to the indirect question can be inferred by combining the answer of direct question with the image. It is difficult to infer the answer to the indirect question only based on the answer to the direct question alone. The explanation is used to train model.
Use the bounding-boxes [x, y, w, h] only to anchor the object in the explanation if it is provided in the given information. We give a expample in the following example.
Expected Output Example "'json { "direct_question": "Has the batter already hit the ball?", "direct_answer": "Yes", "direct_explanation": "The batter's posture [63.74, 84.13, 104.0, 321.12] indicates a completed swing: the arms are fully extended, the bat [562.69, 203.18, 77.31, 22.34] has already passed through the hitting zone, and the ball [78.26, 108.53, 26.41, 26.97] is seen traveling away from the batter. These cues suggest the ball [78.26, 108.53, 26.41, 26.97] was just hit." }
Please give me the output for the data. Captions: ${caption}$ Bounding Boxes: ${boudingbox}$ Output:

## J.2 PROMPTS FOR MODEL TRAINING

---

**Prompt for RL Training**

**Image:** ${image}$

**Prompt:** Please reason step by step, and put your final answer within \boxed{}.

---

**Prompt for Training Data Generation**

**Image:** ${image}$

**Prompt:** You are a visual reasoning question generator. Given:
captions: a list of strings describing the same image bboxes: a list of object entries in the form "¡category¿: [x, y, w, h]" image: the input image contains object bounding boxes
Generate exactly one JSON object with four fields:
"direct_question": string, // a yes/no or simple fact question about visual reasoning in the image "direct_answer": string, // "Yes" or "No" (or a short fact) answering the direct question "direct_explanation": string, // Explain how can you get answer for the direct question, "indirect_question": string, // a multiple-choice question (two options) that is logically implied by the direct question but phrased differently "indirect_answer": string // "optionA" or

"optionB", "indirect_explanation": string, // Explain how can you get answer for the indirect question

Requirements: 1. Both questions must require reasoning about content in the image. 2. The indirect question's correct choice must follow from the direct answer, but the wording/semantic focus should differ (e.g. "Did the batter hit the ball?" → "Is the ball moving toward or away from the batter? optionA: toward optionB: away"). 3. Use the bounding-boxes only to anchor objects in the explanation. 4. Do not invent objects or things not supported by the captions + bboxes. 5. The object in explanations should apear with its coordinates if it appears in the image. 6. The explanation is used to train model. And note that the model will not be provided with captions and the bounding boxes. So please don't generate content like "the captions indicate xxx" or "the bounding boxes xx indicate xxx". 7. Direct questions are yes-or-no questions. Direct questions are asked in a direct way. Indirect questions are asked indirectly, from another angle. The answer to the indirect question can be inferred by combining the answer of direct question with the image. It is difficult to infer the answer to the indirect question only based on the answer to the direct question alone.

If the image is very simple and it is not easy to get a reasoning question. Just output None.

Examples:

Captions: - "A young boy hits a baseball while another boy with an umbrella looks on." - "A little league player is swinging at a ball, and a boy under an umbrella is watching on the sideline."

Bounding Bboxes: - sports ball: [78.26, 108.53, 26.41, 26.97] - person: [63.74, 84.13, 104.0, 321.12] - baseball bat: [562.69, 203.18, 77.31, 22.34] - umbrella: [0.0, 8.33, 223.38, 96.05] - sports ball: [78.26, 108.53, 26.41, 26.97]

Output: "'json { "direct_question": "Has the batter already hit the ball?", "direct_answer": "Yes", "direct_explanation": "The batter's posture [63.74, 84.13, 104.0, 321.12] indicates a completed swing: the arms are fully extended, the bat [562.69, 203.18, 77.31, 22.34] has already passed through the hitting zone, and the ball [78.26, 108.53, 26.41, 26.97] is seen traveling away from the batter. These cues suggest the ball [78.26, 108.53, 26.41, 26.97] was just hit.", "indirect_question": "Is the ball flying toward or away from the batter? optionA: toward optionB: away", "indirect_answer": "optionB", "indirect_explanation": "The ball [78.26, 108.53, 26.41, 26.97] is flying away from the batter [63.74, 84.13, 104.0, 321.12], as the arms are fully extended, the bat [562.69, 203.18, 77.31, 22.34] has already passed through the hitting zone.", }

Captions: - A 70th white birthday cake with flower decorations on top of a table. - A 70th birthday cake sitting on a wooden bench. - A cake is shown on top of a picnic table. - A picnic tables holds a birthday cake while a little girl stands beside. - A picture of a cake on a table.

Bounding Bboxes: - bottle: [194.79, 2.16, 29.74, 76.76] - bottle: [394.79, 2.3, 29.45, 78.22] - bench: [491.29, 114.19, 148.71, 308.01] - bench: [0.0, 118.82, 163.12, 301.3] - knife: [142.13, 112.33, 34.86, 15.82] - cake: [169.84, 166.0, 297.46, 139.14] - person: [0.0, 0.0, 165.26, 318.19] - dining table: [22.12, 32.22, 524.13, 390.45] - book: [240.6, 36.11, 63.43, 13.05] - book: [233.44, 28.77, 66.43, 11.52]

Output: "'json { "direct_question": "Is the number 70 on the cake?", "direct_answer": "Yes", "direct_explanation": "The cake [169.84, 166.0, 297.46, 139.14] show there is a number 70.", "indirect_question": "Is cake for the girl or other one? optionA: the little girl optionB: someone which is not visible in the image", "indirect_answer": "optionB", "indirect_explanation": "The cake [169.84, 166.0, 297.46, 139.14] show there is a number 70. But the little girl [0.0, 0.0, 165.26, 318.19] is very young. Therefore, the cake is for smeoneelse which doesn't appear in the image.", }

Captions: - A 70th white birthday cake with flower decorations on top of a table. - A 70th birthday cake sitting on a wooden bench. - A cake is shown on top of a picnic table. - A picnic tables holds a birthday cake while a little girl stands beside. - A picture of a cake on a table.

Bounding Bboxes: - bottle: [194.79, 2.16, 29.74, 76.76] - bottle: [394.79, 2.3, 29.45, 78.22] - bench: [491.29, 114.19, 148.71, 308.01] - bench: [0.0, 118.82, 163.12, 301.3] - knife: [142.13, 112.33, 34.86, 15.82] - cake: [169.84, 166.0, 297.46, 139.14] - person: [0.0, 0.0,

165.26, 318.19] - dining table: [22.12, 32.22, 524.13, 390.45] - book: [240.6, 36.11, 63.43, 13.05] - book: [233.44, 28.77, 66.43, 11.52]
Output: None
Captions: - "A young boy hits a baseball while another boy with an umbrella looks on." - "A little league player is swinging at a ball, and a boy under an umbrella is watching on the sideline."
Bounding Bboxes: - sports ball: [78.26, 108.53, 26.41, 26.97] - person: [63.74, 84.13, 104.0, 321.12] - baseball bat: [562.69, 203.18, 77.31, 22.34] - umbrella: [0.0, 8.33, 223.38, 96.05] - sports ball: [78.26, 108.53, 26.41, 26.97]
Output: "'json "direct_question": "Has the batter already hit the ball?", "direct_answer": "Yes", "direct_explanation": "The batter's posture [63.74, 84.13, 104.0, 321.12] indicates a completed swing: the arms are fully extended, the bat [562.69, 203.18, 77.31, 22.34] has already passed through the hitting zone, and the ball [78.26, 108.53, 26.41, 26.97] is seen traveling away from the batter. These cues suggest the ball [78.26, 108.53, 26.41, 26.97] was just hit.", "indirect_question": "Is the ball flying toward or away from the batter? optionA: toward optionB: away", "indirect_answer": "optionB", "indirect_explanation": "The ball [78.26, 108.53, 26.41, 26.97] is flying away from the batter [63.74, 84.13, 104.0, 321.12], as the arms are fully extended, the bat [562.69, 203.18, 77.31, 22.34] has already passed through the hitting zone.", Please give me the output for the data. Captions: ${caption}$ Bounding Boxes: ${boudingbox}$ Output:

## K  KEYWORDS FOR IMAGE SELECTION

We list our keywords for image selection as follows.

```
keywords_for_causal_and_intent = ["reaching", "grabbing", "opening",
    ↪ "holding", "looking at", "entering", "because", "after", "fell",
    ↪ "spilled", "dropped"]

keywords_for_complex_counting = ["mirror", "reflected", "reflection",
    ↪ "glass", "see himself", "in front of a mirror"]

keywords_for_time_perceptron_reasoning = ["clock", "watch", "time",
    ↪ "morning", "sunset", "sunrise", "night", "evening"]

keywords_for _action_and_state = ["sport", "game", "match",
    ↪ "competition", "athlete", "player", "team", "stadium", "field",
    ↪ "court", "race", "tournament", "ball", "soccer", "football",
    ↪ "basketball", "baseball", "volleyball", "tennis", "golf",
    ↪ "hockey", "cricket", "running", "jumping", "swimming", "skiing",
    ↪ "skating", "surfing", "cycling", "riding", "throwing", "hitting",
    ↪ "kicking", "diving", "moving"]
```

## L  CLARIFICATION ON THE DIFFERENCE BETWEEN CONVBENCH AND TRAINING DATA

To avoid potential misunderstanding, we explicitly clarify the distinction between the construction of **ConVBench** and the **training data** used for ConVLM. The training data are *entirely auto-generated* by the proposer model: given an image, the proposer produces logically related question-answer pairs, and these pseudo-labels are directly used for weakly supervised RL without any human intervention or manual correction. In contrast, ConVBench is a *curated evaluation benchmark*. This dataset undergoes *human verification* to ensure that each pair of questions is logically valid and that their answers are correct and mutually consistent. No human validation is used for training data, and no ConVBench samples are used during training. Thus, the two pipelines serve different pur-

poses—automatic weak supervision for training versus high-quality, human-validated assessment.

## M   BENCHMARK CHARACTERISTICS ANALYSIS

The performance difference between V*Bench and InfoVQA can be explained by their distinct task emphases. V*Bench primarily evaluates visual reasoning in natural images—covering spatial relations, causal and temporal inference, and multi-step logic—which closely matches the reasoning types emphasized during our consistency-based RL training. In contrast, InfoVQA focuses heavily on OCR, fine-grained text recognition, and structured information extraction, which our training pipeline does not target. Therefore, the discrepancy reflects a task distribution mismatch: ConVLM is optimized for image-grounded reasoning rather than text-centric perception.

## N   COMPARISON WITH PRIOR CONSISTENCY-BASED REASONING METHODS

For completeness, we clarify the conceptual differences between our framework and prior work on consistency in visual or language reasoning. Existing visual reasoning studies (Jing et al., 2022) focus on compositional VQA consistency, such as part–whole or attribute–object relations, and typically implement consistency as deterministic loss constraints. These methods do not consider consistency across distinct but semantically equivalent queries conditioned on the same image. Zhang et al. (2024) introduce another notion of consistency, defined as the agreement between two *different answer formats for the same question* ("What is the answer?" vs. "Is option X correct?"). This type of reasoning consistency evaluates answer-format invariance using an F1 score, and does not involve generating new queries or reasoning over semantic equivalence across questions. In contrast, language-only self-consistency approaches (Wang et al., 2022) operate exclusively during inference via sampling, and do not provide training-based methods.

In contrast, our work introduces a different consistency formulation: *pairwise logical equivalence under the same image*. This setting requires the model to produce coherent answers across logically linked queries conditioned on identical visual evidence, a problem not addressed in previous visual reasoning literature. Moreover, we provide a scalable, fully automatic pipeline for generating large sets of logically equivalent question pairs, enabling reinforcement learning at scale. Finally, consistency is incorporated as an *explicit reward* within the GRPO framework, providing sample- and group-level credit assignment that differs fundamentally from deterministic loss regularization or inference-time sampling heuristics.

## O   HUMAN VERIFICATION OF CONVBENCH

To ensure the validity of the logically equivalent QA pairs in ConVBench, we manually verified all pairs generated by the proposer model (GPT-4.1). In total, the benchmark contains 686 final QA pairs, each derived from independent human inspection. During verification, candidate pairs were categorized as accepted, corrected, or discarded. Among all generated candidates, 389 pairs (57%) were accepted without modification, 150 pairs (22%) required minor corrections, and 147 pairs (21%) were removed due to incorrect or ambiguous logical equivalence.

