# OpenReview forum: "Be Consistent! Enhancing Robust Visual Reasoning in LVLMs with Consistency Constraints"
_ICLR.cc/2026/Conference — Submitted to ICLR 2026_

### Official Review · Reviewer_GjvN · 2025-10-30

**Soundness:** 3
**Presentation:** 2
**Contribution:** 2
**Rating:** 4
**Confidence:** 4

**Summary:**

This paper introduces ConVBench, a benchmark designed to jointly evaluate complex visual reasoning and logical consistency by pairing each image with two logically equivalent questions. To improve consistency in model outputs, the authors propose ConVLM, which uses GRPO-based reinforcement learning with a consistency reward in addition to accuracy supervision.

**Strengths:**

1. The proposed benchmark construction is systematic. The division into six reasoning categories and the human verification pipeline provides a reasonably well-controlled dataset.
2. The proposed method ConVLM outperforms strong open-source baselines and shows non-trivial generalization to external benchmarks.
3. The paper is well-written and ablations are well-conducted to verify the contribution of each component.

**Weaknesses:**

1. Prior works on visual reasoning [1][2][3] have already highlighted the need for enforcing consistency across semantically equivalent prompts. The current method appears to mainly package GRPO + consistency reward + auto-generated pairs, and the conceptual contribution beyond these works is not fully articulated.
2. While the benchmark claims “complex reasoning”, most question types appear to involve single-hop inference rather than multi-step chain-of-thought visual reasoning. It remains unclear whether ConVLM actually improves multi-step visual reasoning, an aspect emphasized in recent VLM reasoning literature.

- [1] Jing, Chenchen, et al. "Maintaining reasoning consistency in compositional visual question answering." CVPR 2022
- [2] Wang, Xuezhi, et al. "Self-Consistency Improves Chain of Thought Reasoning in Language Models." ICLR 2023
- [3] Zhang, Mingyu, et al. "Take a step back: Rethinking the two stages in visual reasoning." ECCV 2024

**Questions:**

1. The authors need to clarify how their notion of reasoning consistency specifically differs from prior studies such as [1][2][3]. It would be helpful to elaborate on the conceptual or methodological distinctions between their formulation and these earlier works.
2. The equivalence between q₁ and q₂ is determined by GPT-4.1, and although human oversight is mentioned, the process lacks quantitative details. To ensure the rationality of the benchmark, the authors should report how many generated pairs were accepted, corrected, or discarded after human verification.
3. The authors are encouraged to conduct a case study on inconsistent reasoning instances to provide more insight. Analyzing when and why inconsistencies persist across different reasoning categories could lead to a deeper and more informative understanding of the model’s behavior.
4. As shown in Table 1, the proposed open-source models outperform other models of similar size but still lag behind large closed-source systems. The authors should discuss whether the proposed method would continue to be effective at larger model scales, or whether such consistency improvements might naturally emerge from increased model capacity and data volume.

---

> ### Author Response · Authors · 2025-11-19
> **Response to Reviewer GjvN**
>
> > Prior works on visual reasoning [1][2][3] have already highlighted the need for enforcing consistency across semantically equivalent prompts. The current method appears to mainly package GRPO + consistency reward + auto-generated pairs, and the conceptual contribution beyond these works is not fully articulated.
> - Thank you for the comment. We fully agree that prior work has discussed consistency in visual reasoning, but our method differs from [1–3] in problem formulation, objective, and mechanism:
>     - Different type of consistency: Prior works [1] focus on compositional VQA consistency (e.g., part–whole, attribute–object relations) or language-only self-consistency. [3] introduces another notion of consistency, defined as the agreement between two different answer formats for the same question (``What is the answer?'' vs. ``Is option X correct?''). This type of reasoning consistency evaluates answer-format invariance. In contrast, we study pairwise logical equivalence under the same image—a setting not explored in prior visual reasoning work.
>     - Consistency as an explicit RL reward: Self-consistency [2] is used only at inference by sampling multiple outputs. Prior visual reasoning consistency methods [1] introduce loss-level constraints. In contrast, our method uses GRPO-based credit assignment, giving sample-wise/group-wise consistency rewards based on cross-query agreement, this leads to very different training method.
>     - Fully automatic consistency-pair generation at scale: None of [1–3] provide a scalable pipeline to auto-generate logically equivalent pairs from raw images. Our proposer model + structured transformation rules enable generating training pairs without human effort, enabling RL on scalable consistency groups.
>
> Thus, the conceptual contribution is not GRPO alone, but the integration of (1) a new pairwise-logical-equivalence consistency formulation, (2) a scalable automatic data generator, and (3) a simple but highly effective consistency reward that produces substantial gains across VLMs. We added the related discussion in our Appendix N.
>
>
> > While the benchmark claims “complex reasoning”, most question types appear to involve single-hop inference rather than multi-step chain-of-thought visual reasoning. It remains unclear whether ConVLM actually improves multi-step visual reasoning, an aspect emphasized in recent VLM reasoning literature.
> - We apologize for the misunderstanding. Our notion of complex reasoning in ConVBench is vision-centric: it targets reasoning that requires integrating multiple visual cues in real-world scenes (spatial layout, temporal dynamics, actions and states, causal relations, and commonsense) and can't be resolved with a blink. This is fundamentally different from multi-step chain-of-thought visual reasoning, which evaluates explicit multi-step inference. ConVBench focuses on multi-cue visual inference rather than CoT-style multi-step reasoning, making the two orthogonal. We revised Appendix A to clarify this point.
>
>
> Questions:
>
> > The equivalence between q₁ and q₂ is determined by GPT-4.1, and although human oversight is mentioned, the process lacks quantitative details. To ensure the rationality of the benchmark, the authors should report how many generated pairs were accepted, corrected, or discarded after human verification.
> - Thank you for the question. We clarify that ConVBench contains 686 logically equivalent QA pairs, and every pair was manually verified. GPT-4.1 only produces candidate pairs; we also provide GPT-4.1 with some ground truth information (bounding boxes, captions) to avoid potential biases and inconsistencies; all final pairs were checked by human annotators. In total, among the 686 generated pairs, 147 pairs (21%) were discarded, 150 pairs (22%) were corrected, and the remaining 389 pairs (57%) were accepted without modification. We added this information in Appendix O.

---

> > ### Author Response · Authors · 2025-11-19
> >
> > > The authors are encouraged to conduct a case study on inconsistent reasoning instances to provide more insight. Analyzing when and why inconsistencies persist across different reasoning categories could lead to a deeper and more informative understanding of the model’s behavior.
> > - Thank you for the helpful suggestion. We agree that a deeper case-study analysis of inconsistent reasoning instances would provide additional insight. Due to space constraints, we were unable to include an expanded qualitative study in the main paper. However, we have added several representative examples to the appendix to illustrate the typical failure patterns across different reasoning categories in Figure 6 of Appendix I, covering five representative inconsistency types across different reasoning categories. We believe these examples illustrate the major failure modes and complement the quantitative results. We also added more analysis in Appendix I. We also show the added analysis as follows:
> > - Cases 2–5 collectively reveal several recurring inconsistency patterns across different reasoning categories. Case 2 shows that intent and action understanding is fragile: small paraphrases around a player’s pose (e.g., “signaling” vs. “ready to catch”) can flip the prediction, indicating unstable grounding from fine-grained body posture to high-level intent. Case 3 highlights similar instability for dynamic outcomes, where a transitional frame (e.g., approaching a base or being tagged) causes the model to disagree on whether a goal state has been reached, even under logically equivalent descriptions. Case 4 exposes temporal reasoning issues: when asked about time-related properties (such as reading a clock or inferring “before/after” relations), the model can produce conflicting answers across equivalent formulations, suggesting it does not maintain a consistent internal representation of temporal state. Case 5 illustrates failures in causal and social reasoning, where different framings of the same situation lead to divergent explanations about motivations or consequences, despite the underlying visual evidence being unchanged. Together, these cases indicate that inconsistencies are most likely to occur when reasoning depends on subtle, multi-cue interpretations of actions, outcomes, time, and social context, underscoring the need for explicit consistency-oriented training.
> >
> >
> > > As shown in Table 1, the proposed open-source models outperform other models of similar size but still lag behind large closed-source systems. The authors should discuss whether such consistency improvements might naturally emerge from increased model capacity and data volume.
> > - Thank you for the question. We agree that larger models tend to exhibit slightly higher consistency. However, scale only partially alleviates inconsistency and does not fundamentally solve it. As shown in Table 1, even very large commercial VLMs such as GPT-4.1, Gemini-2.5-Pro, and Claude-3.7 achieve around 70–75% consistency, meaning they still exhibit approximately 25–30% inconsistency on ConVBench. This indicates that inconsistency persists at scale.
> > - To further clarify this point, we **additionally evaluated large open-source models Qwen2.5-VL-32B and Qwen2.5-VL-72B** under our evaluation protocol. Their consistency scores are 55.20 and 56.09, respectively—only modestly higher than 7B, and still far below our ConVLM-3B, despite being 10×–25× larger. This shows that increased capacity improves consistency only marginally and does not eliminate the issue. We will add these results to our Appendix.
> > - While we would have liked to apply our GRPO-based consistency training to 32B/72B models, this is computationally infeasible for us in terms of limited timelines and GPU resources. In fact, our 7B open-source model ConVLM-7B (73.36% consistency / 66.83% accuracy) already outperforms Claude-3.5-Sonnet-V2 (62.24% / 45.89%) and Claude-3.7-Sonnet (71.17% / 59.13%) on ConVBench, and comes within 3–4 percentage points in consistency of much larger closed-source models such as Gemini-2.5-Pro (74.99%) and GPT-4.1 (76.76%). This suggests that our consistency-based training substantially narrows the gap between small open-source models and powerful proprietary systems on this benchmark.
> > - Therefore, the improvements from our consistency-based RL cannot be attributed to scale alone. Our approach substantially boosts consistency for small models beyond what raw scaling achieves, suggesting that consistency does not naturally emerge solely from larger capacity.

---

> ### Comment · Reviewer_GjvN · 2025-11-28
>
> The authors’ discussion on the clarification of their contribution and extra experiments is clear and addresses my main concerns.  Given the overall contribution and workload of the paper, I am leaning toward borderline accept. I appreciate the authors’ effort and their contribution to the reasoning community.

---

> ### Author Response · Authors · 2025-11-28
>
> We sincerely appreciate your constructive feedback and are happy that our response helped clarify the contribution and address your concerns. Thank you for your valuable comments and for supporting the improvement of our paper.

---

### Official Review · Reviewer_A968 · 2025-11-01

**Soundness:** 3
**Presentation:** 3
**Contribution:** 3
**Rating:** 6
**Confidence:** 3

**Summary:**

This paper addresses the consistency issue in vision-language models (VLMs), where a model may provide conflicting answers to two logically equivalent questions. For instance, when asked "Is object X moving away?" and "What is the action of object X? Options: A) toward, B) away," a consistent VLM should output aligned responses such as ("yes" and "away") or ("no" and "toward"). However, existing VLMs often fail to maintain such logical alignment.

To tackle this problem, they introduce ConVBench, a novel benchmark where each image is paired with two logically equivalent 〈Question, Answer〉 pairs. They also propose two evaluation metrics: standard accuracy and a consistency score, which measures whether a model correctly answers both equivalent questions simultaneously.

Furthermore, they present ConVLM, a GRPO-based reinforcement learning framework designed to enhance VLM consistency. The reward function incorporates both accuracy and consistency objectives. Training data is automatically generated by a proposer agent (GPT-4), which produces logically equivalent 〈Q, A〉 pairs based on MSCOCO images. Extensive experiments on ConVBench and the V* benchmark demonstrate the effectiveness of ConVLM.

**Strengths:**

1. This paper proposes ConVBench, a novel benchmark for evaluating the consistency of Vision-Language Models (VLMs) using logically equivalent question-answer pairs. In addition to conventional accuracy metrics, ConVBench introduces a dedicated consistency score to assess whether a model provides coherent answers across reformulated versions of the same question.

2. To address the inconsistency issue, the authors introduce a GRPO-based reinforcement learning framework that incorporates both accuracy and consistency rewards in its objective function. Experimental results demonstrate that the resulting model, ConVLM, achieves notable improvements on both ConVBench and the V* benchmark.

**Weaknesses:**

1.The details of ConVBench's construction are insufficiently described in Section 2. It remains unclear whether its generation process aligns with the training-set generation pipeline illustrated by the proposer in Figure 3. A more thorough explanation of the benchmark creation methodology, including data sources, transformation rules, and validation procedures, is necessary to ensure reproducibility and proper interpretation.

2.Table 1 indicates that commercial VLMs (e.g., Claude, Gemini, ChatGPT) achieve relatively high consistency scores. This raises the question of whether scale (i.e., larger parameter counts) inherently mitigates consistency issues. It would be insightful to include results from large open-source models such as Qwen2.5-VL-72B to better disentangle the impact of model size from architectural or training data factors.

3 According to Table 3, most evaluated models perform better on InfoVQA than on V, except for ConVLM. This discrepancy may stem from domain relevance: if ConVLM is trained on MSCOCO-derived data, does its relatively lower performance on InfoVQA indicate a domain adaptation gap, while its strong result on V reflects better alignment with the MSCOCO distribution? Further analysis on the domain characteristics of each benchmark would help contextualize these results.

**Questions:**

1. How is ConVBench generated?

2. Does the consistency issue persist in large-scale VLMs?

3. Why does InfoVQA not benefit as much as V*?

---

> ### Author Response · Authors · 2025-11-19
> **Response to Reviewer A968**
>
> > 1.The details of ConVBench's construction are insufficiently described in Section 2. It remains unclear whether its generation process aligns with the training-set generation pipeline illustrated by the proposer in Figure 3. A more thorough explanation of the benchmark creation methodology, including data sources, transformation rules, and validation procedures, is necessary to ensure reproducibility and proper interpretation.
> - Thank you for the comment, and we apologize for the confusion. We would like to clarify that the construction of ConVBench (including data sources for each sub-type, transformation/generation rules, and human validation procedures) is already described in the paper (Sec. 2 and Appendix J.1). ConVBench and the training-set generation pipeline share similar logical transformation principles, but they are not identical: the training data is generated fully automatically by the proposer without human intervention, whereas ConVBench includes an additional human verification step solely to ensure evaluation quality, and specilized prompts for different sub-sets. To improve clarity, we additionlly added the clarification for the difference between ConVBench and the training set in Appendix L.
>
>
> > 2.Table 1 indicates that commercial VLMs (e.g., Claude, Gemini, ChatGPT) achieve relatively high consistency scores. This raises the question of whether scale (i.e., larger parameter counts) inherently mitigates consistency issues. It would be insightful to include results from large open-source models such as Qwen2.5-VL-72B to better disentangle the impact of model size from architectural or training data factors.
> - Thank you for the suggestion. We agree that larger models generally exhibit slightly better consistency—a trend broadly observed across VLM literature. However, scaling alleviates inconsistency only marginally and does not fundamentally resolve it. While Table 1 shows that large commercial VLMs (e.g., Claude, Gemini) reach higher consistency, scale alone is insufficient: even GPT-4.1 and Claude-3.7 still exhibit around 25–30% inconsistency on ConVBench. Likewise, Qwen2.5-VL-7B, despite being substantially larger than our 3B base, remains far below ConVLM-3B.
> - To further disentangle the effect of scale, we additionally evaluated **Qwen2.5-VL-32B and Qwen2.5-VL-72B under our evaluation protocol**. Their performance is: Qwen2.5-VL-32B: Consistency 55.20, Accuracy 33.98 and Qwen2.5-VL-72B: Consistency 56.09, Accuracy 34.21. Although these large models perform somewhat better than 7B, they still retain a substantial inconsistency gap and remain far below our ConVLM-3B, demonstrating that scale improves but does not solve consistency challenge. We will add these results in our Appendix.
> - While we would have liked to apply our GRPO-based consistency training to 32B/72B models, this is computationally infeasible for us in terms of limited timelines and GPU resources. In fact, our 7B open-source model ConVLM-7B (73.36% consistency / 66.83% accuracy) already outperforms Claude-3.5-Sonnet-V2 (62.24% / 45.89%) and Claude-3.7-Sonnet (71.17% / 59.13%) on ConVBench, and comes within 3–4 percentage points in consistency of much larger closed-source models such as Gemini-2.5-Pro (74.99%) and GPT-4.1 (76.76%). This suggests that our consistency-based training substantially narrows the gap between small open-source models and powerful proprietary systems on this benchmark.
> - In summary, the current results already show that even very large models remain inconsistent, and our method improves a 3B/7B model beyond the raw performance of much larger backbones.
>
>
>
> > 3 According to Table 3, most evaluated models perform better on InfoVQA than on V, except for ConVLM. This discrepancy may stem from domain relevance: if ConVLM is trained on MSCOCO-derived data, does its relatively lower performance on InfoVQA indicate a domain adaptation gap, while its strong result on V reflects better alignment with the MSCOCO distribution? Further analysis on the domain characteristics of each benchmark would help contextualize these results.
> - Thank you for the insightful question. The difference between V*Bench and InfoVQA arises mainly from their task characteristics. V*Bench focuses on visual reasoning over the real-world scenes, including multi-step reasoning, spatial relations, temporal understanding, and other forms of image-grounded logic—categories that align closely with the reasoning types used in our consistency-based RL training. In contrast, InfoVQA heavily emphasizes OCR, fine-grained text recognition, and structured information extraction, which our training pipeline does not target. Therefore, the relative gap reflects a task distribution mismatch: our method boosts visual reasoning robustness, not text-centric perception skills. We have now added this in Appendix M.

---

### Official Review · Reviewer_rAAu · 2025-11-02

**Soundness:** 3
**Presentation:** 3
**Contribution:** 3
**Rating:** 4
**Confidence:** 4

**Summary:**

This paper addresses the critical yet under-explored issue of logical consistency in Large Vision-Language Models (LVLMs). The authors argue that while LVLMs have strong perceptual abilities, they often fail at complex visual reasoning tasks and produce contradictory answers to logically equivalent questions. To tackle this, the paper presents two main contributions:

1. ConVBench: A new vision-centric benchmark designed to rigorously evaluate both complex reasoning and logical consistency. Each image in ConVBench is paired with two logically equivalent questions across six reasoning categories. The benchmark introduces two novel metrics: logical consistency and robust accuracy.

2. ConVLM: A framework for improving LVLM reasoning by enforcing consistency. The method uses GRPO with a novel dual-reward mechanism. This reward combines a standard accuracy signal with a new consistency reward, which encourages the model to produce agreeing outputs for logically equivalent question pairs. Notably, the training data for this process is generated automatically by a powerful LVLM  and then validated by humans, making the approach scalable.

The authors demonstrate through extensive experiments that their ConVLM-7B model achieves state-of-the-art results among open-source models on ConVBench, significantly outperforming strong baselines. Furthermore, the model shows excellent generalization capabilities on other challenging benchmarks like V*Bench and InfoVQA, indicating that the consistency training imparts a more robust and generalizable reasoning ability.

**Strengths:**

1. The paper tackles a fundamental flaw in current LVLMs. Logical consistency is a cornerstone of reliable and trustworthy AI, and this work provides a formal framework to measure and improve it.

2. ConVBench is a valuable asset for the field. Its design principles—focusing on vision-centric tasks, complex reasoning, and logically equivalent question pairs—fill an important gap in existing evaluation suites.

3. The ConVLM framework is elegant and well-motivated. The key novelty lies in the dual-reward design that explicitly optimizes for consistency alongside accuracy.

**Weaknesses:**

1. While the concept is powerful, the examples shown primarily involve rephrasing or direct one-step implications (e.g., hitting a ball implies it's moving away). The paper could benefit from a more detailed discussion on the diversity and complexity of the logical relationships present in ConVBench.

2. Proving the method's efficacy on STEM-related data, such as visual mathematics or physics problems (e.g., on benchmarks like MathVista), would greatly strengthen the paper's claims.

**Questions:**

The ablation study in Table 2 shows that training with only the consistency reward (w/o-Acc) leads to a dramatic improvement in both consistency and accuracy over the baseline. This is a fascinating result. Could you elaborate on why enforcing consistency provides such a strong implicit signal for accuracy?

---

> ### Author Response · Authors · 2025-11-19
> **Response to Reviewer rAAu**
>
> > While the concept is powerful, the examples shown primarily involve rephrasing or direct one-step implications (e.g., hitting a ball implies it's moving away). The paper could benefit from a more detailed discussion on the diversity and complexity of the logical relationships present in ConVBench.
> - Thank you for the suggestion. We agree with your comment that the consistent semantic relation in the mentioned example can indeed be reasoned through direct implication. However, our intention was not to emphasize the complexity of logical relationship between questions, but rather to highlight the need for multi-cue visual reasoning and the model's weakness on consistency reasoning. For example, in Figure 6 (Case 4), the model must jointly interpret the clock and lighting conditions in the image to answer correctly. Moreover, as shown in Figure 6, even for seemingly simple logical-consistency cases, existing LVLMs still struggle to produce satisfactory results.
> - In response to your valuable comment, we have now added more detailed discussion and hope this adds more information for you. In particular, we categorize typical failure modes into: (i) high consistency but low accuracy, where the model is confidently wrong because it stably misinterprets subtle visual cues (e.g., transitional action states or ambiguous spatial layouts); and (ii) high accuracy but low consistency, where the model answers correctly for one formulation but flips under paraphrases or equivalent descriptions, revealing sensitivity to linguistic framing rather than visual evidence. Across cases 1–5, we observe recurring patterns in action/intent, dynamic outcome, temporal, and causal/social reasoning, where small changes in wording or fine-grained visual details lead to instability. Please refer to Appendix I for the detailed analysis.
>
>
> > Proving the method's efficacy on STEM-related data, such as visual mathematics or physics problems (e.g., on benchmarks like MathVista), would greatly strengthen the paper's claims.
> - Thank you for the suggestion. STEM-oriented benchmarks such as MathVista are indeed valuable, but they focus on a fundamentally different reasoning paradigm—symbolic mathematical manipulation, diagram-based deduction, and OCR-heavy perception. Our work, in contrast, targets vision-centric reasoning grounded in natural images (spatial layout, actions, temporal cues, causal inference, commonsense). Therefore, we did not include MathVista in the main paper. However, to address the reviewer’s concern, we conducted an evaluation on MathVista’s test_miniset during the rebuttal period. We observe that ConVLM-3B achieves an accuracy of 0.641, compared to 0.604 for the base Qwen2.5-VL-3B, indicating that our consistency-based training does not harm symbolic/STEM reasoning and may provide slight benefits. We will add these results in our Appendix.
>
>
> Questions:
> >The ablation study in Table 2 shows that training with only the consistency reward (w/o-Acc) leads to a dramatic improvement in both consistency and accuracy over the baseline. This is a fascinating result. Could you elaborate on why enforcing consistency provides such a strong implicit signal for accuracy?
> - Thank you for highlighting this phenomenon. We agree that the result is interesting. Enforcing consistency across logically equivalent questions provides a much stronger training signal than it may appear. First, consistency encourages the model to form stable and image-grounded representations: to answer two logically linked queries coherently, the model cannot rely on superficial cues and must reduce random fluctuations in its reasoning. Empirically, many pre-RL errors are “unstable” (the model answers differently under paraphrases), and reducing this instability naturally improves correctness even without accuracy labels.
> - Under GRPO, the consistency reward provides a much stronger signal than simple string matching: it enforces group-level stability across multiple rollouts. Rollouts that remain coherent across logically equivalent queries receive higher relative advantages, while inconsistent ones are penalized. This acts as a regularizer that reduces variance, increases confidence, and encourages the model to rely on more stable, image-grounded reasoning. As a result, accuracy also improves—even without explicit accuracy supervision.

---

### Official Review · Reviewer_dW82 · 2025-11-03

**Soundness:** 2
**Presentation:** 2
**Contribution:** 1
**Rating:** 2
**Confidence:** 5

**Summary:**

This paper introduces ConVBench, a novel benchmark designed to evaluate both complex visual reasoning and logical consistency in Large Vision-Language Models (LVLMs) using logically equivalent question-answer pairs. To address the limitations of existing models, the authors propose ConVLM, a weakly supervised framework that enhances robust visual reasoning through Group Relative Policy Optimization (GRPO) with a novel consistency-based reward. ConVLM achieves SOTA performance on ConVBench and demonstrates strong generalization to other benchmarks. The work highlights the importance of consistency in visual reasoning and offers a new approach to training more robust LVLMs.

**Strengths:**

- The ConVBench addresses the crucial aspects of complex vision-centric reasoning and logical consistency, which are often overlooked in existing benchmarks. The design with logically equivalent question pairs provides a robust mechanism for assessing reasoning consistency.
- The proposed ConVLM framework, which integrates GRPO with a novel dual reward mechanism (accuracy and consistency), is innovative. The ability to automatically generate question-answer pairs and optimize models without strict answer supervision makes the approach scalable and efficient.
- The strong generalization performance of ConVLM to external visual reasoning benchmarks (V*Bench, InfoVQA-test) is a significant strength.

**Weaknesses:**

- While the paper makes a valuable step by focusing on pairwise answer consistency for question pairs, it does not model cross-image consistency. This limits the evaluation of consistency to isolated instances rather than broader contextual or temporal coherence across related visual information.
- The reliance on a Large Language Model (GPT-4.1) for automatically generating logically equivalent question-answer pairs introduces a potential dependency and scalability challenge. If the underlying LLM itself exhibits biases or inconsistencies, it could subtly affect the quality and diversity of the generated training data, even with human validation.
- The paper describes ConVLM as a "weakly supervised framework", but the pipeline involves generating "pseudo-answers" and human validation. This blend of automated generation and human curation makes the "weakly supervised" claim somewhat ambiguous.
- While the dual reward mechanism (accuracy and consistency) is effective, the consistency reward is based solely on string matching. This might be too simplistic for capturing nuanced semantic consistency. More sophisticated methods for evaluating logical consistency beyond exact string matches, such as semantic similarity metrics or entailment checks, could potentially offer richer and more robust training signals.

**Questions:**

- The prompt templates for question generation are detailed in Appendix J. Could the authors discuss the sensitivity of the generated questions and pseudo-answers to prompt engineering? Were different prompt variations explored, and how robust is the data generation process to changes in the prompts?
- Figure 6 provides a case study of incorrect answers. Could the authors provide a more detailed qualitative analysis of the types of errors ConVLM makes compared to baselines, especially focusing on cases where it achieves high consistency but low accuracy, or vice-versa?

---

> ### Author Response · Authors · 2025-11-19
> **Response to Reviewer dW82**
>
> > While the paper makes a valuable step by focusing on pairwise answer consistency for question pairs, it does not model cross-image consistency. This limits the evaluation of consistency to isolated instances rather than broader contextual or temporal coherence across related visual information.
> - We would like to clarify that this aspect is explicitly acknowledged and discussed in our paper (Appendix D, “Future Work”). Our work intentionally focuses on pairwise answer consistency for logically equivalent questions conditioned on a single image, as a first step toward consistency-based reward design.
> - Modeling cross-image consistency introduces substantial additional challenges—for example, defining semantic equivalence across images, constructing reliable cross-image (question) pairs, and handling visual variations—each of which requires dedicated methodological development and dataset construction. These issues are orthogonal to the core contribution of our work: demonstrating that intra-image logical consistency reward can significantly improve robustness in visual reasoning.
> - We agree that extending the consistency reward to cross-image scenarios is a valuable direction, and we already highlight it as an important avenue for future exploration.
>
> >  The reliance on a Large Language Model (GPT-4.1) for automatically generating logically equivalent question-answer pairs introduces a potential dependency and scalability challenge. If the underlying LLM itself exhibits biases or inconsistencies, it could subtly affect the quality and diversity of the generated training data, even with human validation.
> - We thank the reviewer for the comment. GPT-4.1 is used only as a proposer to draft question–answer pairs; we also provide GPT-4.1 with some ground truth information (bounding boxes, captions) to avoid potential biases and inconsistencies; all ConVBench instances are manually validated, ensuring that the benchmark itself does not inherit GPT-4.1's biases. For training, our framework is explicitly designed to be robust to noisy or biased pseudo-answers: the consistency reward does not rely on strict correctness, and ablations show that even with imperfect proposer labels, the model still improves substantially.
> - Importantly, the strong gains of ConVLM on independent benchmarks such as V*Bench and InfoVQA-test—datasets not generated using GPT-4.1—suggest that the model does not simply absorb proposer-specific artifacts, but learns generalizable visual reasoning behavior. Moreover, the proposer in our framework is model-agnostic and can be replaced by cheaper or domain-specific LVLMs.
>
> >  The paper describes ConVLM as a "weakly supervised framework", but the pipeline involves generating "pseudo-answers" and human validation. This blend of automated generation and human curation makes the "weakly supervised" claim somewhat ambiguous.
> - Thank you for the comment. In our framework, “weakly supervised” refers specifically to the training stage of ConVLM, where the model is optimized using automatically generated question pairs and pseudo-answers **without any human annotations**. The human validation mentioned in the paper applies only to ConVBench, our evaluation benchmark, to ensure high-quality test data—not to the training pipeline. During training, ConVLM relies solely on proposer-generated data and the consistency/accuracy rewards, which fit the standard definition of weak supervision. We will clarify this distinction to avoid ambiguity.

---

> > ### Author Response · Authors · 2025-11-19
> >
> > >  While the dual reward mechanism (accuracy and consistency) is effective, the consistency reward is based solely on string matching. This might be too simplistic for capturing nuanced semantic consistency. More sophisticated methods for evaluating logical consistency beyond exact string matches, such as semantic similarity metrics or entailment checks, could potentially offer richer and more robust training signals.
> > - Thank you for the suggestion. Our choice of a simple string-based consistency reward is deliberate: it provides a reliable, low-noise signal during RL, and our results show that even this minimal form already yields strong gains. We fully agree that more sophisticated semantic or entailment-based consistency measures could offer richer supervision, as noted in our Future Work. Our framework is agnostic to the specific consistency function, and such advanced evaluators can be integrated seamlessly.
> > - In response to the reviewer’s comment, in the rebuttal term, we also experimented with a multimodal consistency evaluator using a lightweight LVLM (LLaVA-1.5-7B). Given the image, question, and two generated answers, the model assesses whether the answers are semantically compatible with each other and the visual content. While this provides a more expressive signal than string matching, we found that it achieves performance comparable to the simple reward (e.g., improving Qwen2.5-VL-3B from 41.89%/33.23% to 70.23%/64.89% in terms of consistency and accuracy on ConVBench), but at a substantially higher computational cost—each RL rollout requires additional LVLM inference, significantly slowing training. Due to the minimal empirical advantage but much higher cost, we retain the simpler consistency reward in this work.
> >
> >
> > Questions:
> > > The prompt templates for question generation are detailed in Appendix J. Could the authors discuss the sensitivity of the generated questions and pseudo-answers to prompt engineering? Were different prompt variations explored, and how robust is the data generation process to changes in the prompts?
> > - We thank the reviewer for the question. We explored several prompt variations (e.g., phrasing changes, different levels of detail, and alternative constraint wording) and observed that the generated question pairs were largely stable across prompts. Minor differences do not affect our pipeline because all ConVBench instances undergo human validation, and the RL training is inherently robust to noise in pseudo-answers. We will clarify this robustness and describe the prompt variants tested in the revision.
> >
> > > Figure 6 provides a case study of incorrect answers. Could the authors provide a more detailed qualitative analysis of the types of errors ConVLM makes compared to baselines, especially focusing on cases where it achieves high consistency but low accuracy, or vice-versa?
> > - Thank you for the suggestion. Figure 6 already provides a case study of incorrect predictions. In particular, we categorize typical failure modes into: (i) high consistency but low accuracy, where the model is confidently wrong because it stably misinterprets subtle visual cues (e.g., transitional action states or ambiguous spatial layouts); and (ii) high accuracy but low consistency, where the model answers correctly for one formulation but flips under paraphrases or equivalent descriptions, revealing sensitivity to linguistic framing rather than visual evidence. Across cases 1–5, we observe recurring patterns in action/intent, dynamic outcome, temporal, and causal/social reasoning, where small changes in wording or fine-grained visual details lead to instability. Please refer to Appendix I for the detailed analysis.

---

### Author Response · Authors · 2025-11-24

Dear Reviewers, thank you for your time and comments. We have carefully addressed all of the raised concerns in our rebuttal, and we kindly invite you to reconsider your evaluation in light of the clarifications provided.

---

### Meta-Review · Area_Chair_G4Ky · 2025-12-27

**Summary:**

The paper introduces ConVBench, a novel vision-centric benchmark designed to evaluate both complex visual reasoning and logical consistency in Large Vision-Language Models (LVLMs) by pairing each image with two logically equivalent questions across six categories. It also proposes ConVLM, a weakly supervised framework that leverages Group Relative Policy Optimization (GRPO)-based reinforcement learning with a dual reward mechanism (accuracy and consistency) to enhance robust visual reasoning. ConVLM achieves state-of-the-art performance on ConVBench among open-source models and demonstrates strong generalization to external benchmarks like V*Bench and InfoVQA-test.

Reviewers’ core concerns informing the decision included:
* The missing modeling of cross-image consistency and reliance on GPT-4.1 for data generation (Reviewer dW82);
* Efficacy on STEM-related data and diversity of logical relationships in ConVBench (Reviewer rAAu);
* ConVBench construction details and consistency issue in large-scale VLMs (Reviewer A968);

Overall, reviewers have negative scores before the discussion, and the core concerns still stand following the rebuttal. Therefore, the reviewing panel is inclined to recommend a major revision of the paper before it can be considered for acceptance.

**Reviewer Concerns:**

The concerns on the missing modeling of cross-image consistency (Reviewer dW82), and the efficacy on STEM-related data and diversity of logical relationships in ConVBench (Reviewer rAAu) are still outstanding post-rebuttal.

**Reviewer Scores:**

Given the concerns from Reviewer dW82, and rAAu are not fully addressed by the authors’ response, the chance for them to increase their scores is relatively low.

---

### Decision · Program_Chairs · 2026-01-26

Reject